# ASOR: Anchor State Oriented Regularization for Policy Optimization under Dynamics Shift

## Abstract

To train neural policies in environments with diverse dynamics, Imitation from Observation (IfO) approaches aim at recovering expert state trajectories. Their success is built upon the assumption that the stationary state distributions induced by optimal policies remain similar despite dynamics shift. However, such an assumption does not hold in many real world scenarios, especially when certain states become inaccessible during environment dynamics change. In this paper, we propose the concept of anchor states which appear in all optimal trajectories under dynamics shift, thereby maintaining consistent state accessibility. Instead of direct imitation, we incorporate anchor state distributions into policy regularization to mitigate the issue of inaccessible states, leading to the ASOR algorithm. By formally characterizing the difference of state accessibility under dynamics shift, we show that the anchor state-based regularization approach provides strong lower-bound performance guarantees for efficient policy optimization. We perform extensive experiments across various online and offline RL benchmarks, including Gridworld, MuJoCo, MetaDrive, D4RL, and a fall-guys like game environment, featuring multiple sources of dynamics shift. Experimental results indicate ASOR can be effectively integrated with several state-of-the-art cross-domain policy transfer algorithms, substantially enhancing their performance.

## 1 Introduction

Recent data-driven Reinforcement Learning (RL) (Sutton & Barto, 1998) approaches facilitate efficient and large-scale policy optimization using either a static dataset (Wu et al., 2019; Fujimoto et al., 2019) or a trajectory buffer updated during training (Lillicrap et al., 2016; Haarnoja et al., 2018). However, these methods generally assume that the data is sampled from a single static environment with constant state transition probabilities. This is usually not the case in real-world applications where environment dynamics can vary a lot, i.e., in environments with *dynamics shift*. For instance, the recommender agent on a short-video platform needs to adapt to time-varying and heterogeneous user preferences (Xue et al., 2022; 2023b). An embodied robotic agent may operate in environments with distinct morphologies and joint torque (Liu et al., 2022). In such tasks, achieving effective RL policies necessitates extensive training data with sufficient dynamics coverage (Liu et al., 2022; Li et al., 2023), and the training process is most likely to be unstable (Luo et al., 2022; Xue et al., 2023a). Therefore, a critical question arises: *how can RL policies be efficiently optimized using data collected under dynamics shift?*

In recent years, considerable research efforts have been devoted to cross-dynamics policy training. To mitigate the problem of inefficient data usage, Imitation from Observation (IfO) algorithms (Wu et al., 2019; Torabi et al., 2018b; Jiang et al., 2020) aim at recovering state trajectories of expert demonstrations. Assuming the optimal state trajectories to be similar across different dynamics, IfO can learn from data with dynamics shift (Gangwani & Peng, 2020; Desai et al., 2020; Radosavovic et al., 2021) because expert state trajectories in one domain can be informative in other domains. However, such assumption will not hold in many real world scenarios, especially under the variation of state accessibility, where certain states are no longer accessible during environment dynamics change. For example, an autonomous vehicle might drive through intersections safely at high speed under low traffic densities, but will face the risk of crashing into other vehicles under high traffic densities. Therefore, states representing "safe driving at high speed" are inaccessible in certain dynamics, leading to distinct stationary state distributions. In such cases, expert trajectories with dynamics shift can be misleading.

To deal with the issue of different state distributions, reference states with evolving accessibility should be excluded during training. We define *anchor states* which appear in all optimal trajectories, maintaining the same accessibility across different dynamics. To learn from anchor state distributions, IfO approaches directly perform distribution matching, but they cannot be naturally integrated with datasets including reward signals and require the demonstrations to be optimal. Instead, we employ anchor states for policy regularization based on the standard RL objective. The resulting constrained policy optimization (CPO) (Achiam et al., 2017) problem requires the policy not only to optimize the expected policy return, but also to generate a stationary state distribution close to the anchor state distribution. By formally characterizing the difference of state accessibility under dynamics shift, we manage to derive strong lower-bound performance guarantees for such a policy regularization procedure. The analysis is built on a weaker assumption than previous works. In practice, non-anchor states tend to have unreliable demonstrations, so policies are also encouraged to generate distinct state distributions on these states. Simplifying the CPO problem with Lagrangian multipliers, the policy regularization can be realized by a simple reward augmentation with state density ratios.

Summarizing these ideas, we propose ASOR (**A**nchor **S**tate **O**riented **R**egularization), a reward augmentation algorithm which can be a general add-on module to existing cross-dynamics RL algorithms (Chen et al., 2021; Luo et al., 2022). In empirical evaluations, we consider the toy environment Minigrid (Chevalier-Boisvert et al., 2023), simulated robotics environment MuJoCo (Todorov et al., 2012), simulated autonomous-driving environment MetaDrive (Li et al., 2023), and a large-scale fall guys-like game environment. The tasks include both online and offline RL setting and involve multiple sources of dynamics shift including obstacle layout, traffic density, body mass, joint damping, and wind speed. Our contributions in this paper can be summarized as follows: 1) By restricting policy regularization only to anchor states, we alleviate the issue of evolving optimal state distributions and propose the ASOR algorithm. 2) We derive strong lower-bound performance guarantees for anchor state-based policy regularization. 3) We apply ASOR to extensive benchmark environments with both online and offline RL and various sources of environment dynamics shift, where ASOR exhibits superior performance when combined with multiple state-of-the-art algorithms.

## 2 BACKGROUND

### 2.1 PRELIMINARIES

To model a set of decision-making tasks with different environment dynamics, we consider the Hidden Parameter Markov Decision Process (HiP-MDP) (Doshi-Velez & Konidaris, 2016) defined by a tuple $(\mathcal{S}, \mathcal{A}, \Theta, T, r, \gamma, \rho_0)$, where $\mathcal{S}$ is the state space and $\mathcal{A}$ is the bounded action space with actions $a \in [-1, 1]$. $\Theta$ is the space of hidden parameters. $T_\theta(s'|s, a)$ is the transition function conditioned on $(s, a)$, as well as a hidden parameter $\theta$ sampled from $\Theta$. $r(s, a, s')$ is the environment reward function. By taking all $s, a, s'$ into account, the reward function inherently includes the transition information and does not change in different dynamics. We also assume $r(s, a, s')$ w.r.t. the action $a$ is $\lambda$-Lipschitz. Discussions on these Lipschitz properties can be found in Appendix A.3. $s'$ is termed as *accessible* from $s$ under dynamics $T^1$ if $\sum_{a \in \mathcal{A}} T(s'|s, a) > 0$. $\gamma \in (0, 1)$ is the discount factor and $\rho_0(s)$ is the initial state distribution.

Policy optimization under dynamics shift aims at finding the optimal policy that maximizes the expected return under all possible $\theta \in \Theta$: $\pi^* = \arg \max_\pi \eta(\pi) = \mathbb{E}_\theta \mathbb{E}_{\pi, T_\theta} \left[ \sum_{t=0}^{\infty} \gamma^t r(s_t, a_t, s_{t+1}) \right]$, where the expectation is under $s_0 \sim \rho_0$, $a_t \sim \pi(\cdot|s_t)$, and $s_{t+1} \sim T_\theta(\cdot|s_t, a_t)$. The Q-value $Q_T^\pi(s, a)$ denotes the expected return after taking action $a$ at state $s$: $Q_T^\pi(s, a) = E_{\pi, T} \left[ \sum_{t=0}^{\infty} \gamma^t r(s_t, a_t, s_{t+1}) | s_0 = s, a_0 = a \right]$. The value function is defined as $V_T^\pi(s) = \mathbb{E}_{a \sim \pi(\cdot|s)} Q_T^\pi(s, a)$. The optimal policy $\pi^*$ under $T$ is defined as $\pi_T^* = \arg \max_\pi \mathbb{E}_{s \sim \rho_0} V_T^\pi(s)$. We also intensely use the stationary state distribution (also referred to as the state occupation function) $d_T^\pi(s) = (1 - \gamma) \sum_{t=0}^{\infty} \gamma^t p(s_t = s | \pi, T)$. The stationary state distribution under the optimal policy is denoted as $d_T^*(s)$, which is the shorthand for $d_T^{\pi_T^*}(s)$. $d_T^*(s)$ will be briefly termed as optimal state distribution in the rest of this paper.

### 2.2 RELATED WORK

**Cross-domain Policy Transfer** Cross-domain policy transfer (Niu et al., 2024) focuses on training policies in source domains and testing them in the target domain. In this paper, we focus on a

---

[1] $T$ without subscript $\theta$ refers to the transition function under any of the hidden parameter $\theta$.

related problem of efficient training in multiple source domains. The resulting algorithm can be combined with any of the following cross-domain policy transfer algorithms to improve the test-time performance. In online RL, VariBAD (Zintgraf et al., 2020) trains a context encoder with variational inference and trajectory likelihood maximization. CaDM (Lee et al., 2020) and ESCP (Luo et al., 2022) construct auxiliary tasks including next state prediction and contrastive learning to train the encoders. Instead of relying on context encoders, DARC (Eysenbach et al., 2021) makes domain adaptation by assigning higher rewards on samples that are more likely to happen in the target environment. Encoder-based (Chen et al., 2021) and reward-based (Liu et al., 2022) policy transfer algorithms are also effective in offline policy adaptation and have been extended to offline-to-online tasks (Niu et al., 2022; 2023). VGDF (Xu et al., 2023) use ensembled value estimations to perform prioritized Q-value updates, which can be applied in both online and offline settings. SRPO (Xue et al., 2023a) focus on a similar setting of efficient data usage with this paper, but is based on a strong assumption of universal identical state accessibility. We demonstrate that such an assumption will not hold in many tasks and a more delicate characterization of state accessibility will lead to better theoretical and empirical results.

**Imitation Learning from Observations** Imitation Learning from Observation (IfO) approaches obviate the need of imitating expert actions and is suitable for tasks where action demonstrations may be unavailable. BCO (Wu et al., 2019) and GAIfO (Torabi et al., 2018b) are two natural modifications of traditional Imitation Learning (IL) methods (Ho & Ermon, 2016) with the idea of IfO. IfO has also been found promising when the demonstrations are collected from several environments with different dynamics. Usually an inverse dynamics model is first trained with samples from the target environment by supervised learning (Wu et al., 2019; Radosavovic et al., 2021), variational inference (Liu et al., 2020), or distribution matching (Desai et al., 2020). It is then used to recover the adapted actions in samples from the source environment. The recovered samples can be used to update policies with action discrepancy loss (Gangwani & Peng, 2020; Radosavovic et al., 2021; Liu et al., 2020). To our best knowledge, only HIDIL (Jiang et al., 2020) considered state distribution mismatch across different dynamics, where policies are allowed to take extra steps to reach the next state specified in the expert demonstration. We consider in this paper a more general setting where states in expert demonstrations may even be inaccessible.

## 3 ANCHOR STATE ORIENTED POLICY REGULARIZATION

In this section, we first provide a motivating example in Sec. 3.1 to demonstrate that the assumption of identical state distribution under dynamics shift may not hold in certain scenarios. Then in Sec. 3.2 we propose the approach of anchor state oriented policy regularization that do not rely on this assumption. The approach involves reward augmentations with logarithms of density ratios, but the density ratios are intractable to compute. We therefore propose a data-based method to estimate the ratios in Sec. 3.3 and conclude the section with the practical algorithm procedure in Sec. 3.4.

### 3.1 MOTIVATING EXAMPLE

Previous state-only policy transfer algorithms are based on the assumption that the optimal state distribution $d^*(s)$ remains the same under different environment dynamics, either implicitly (Desai et al., 2020; Jiang et al., 2020) or explicitly (Xue et al., 2023a). Fig. 1 demonstrates an example lava world task with dynamics shift where such assumption does not hold. We consider a 3-dimensional state space including agent row, agent column, and a 0-1 variable indicating whether there is an accessible lava block near the agent. The agent starts from the blue grid and targets at the green grid with positive reward. It also receives a small negative reward on each step. The red grid stands for the dangerous lava area which ends the trajectory on agent entering. One lava block is fixed at Row 1, while the other may appear at Row 2, 3, 4, and 5. Fig. 1 demonstrates two examples where the movable lava block is at Row 2 and Row 4. Taking state (1,3,0) as an example, the same action of "moving down" gives rise to different next state distributions due to distinct lava positions, leading to environment dynamics shift.

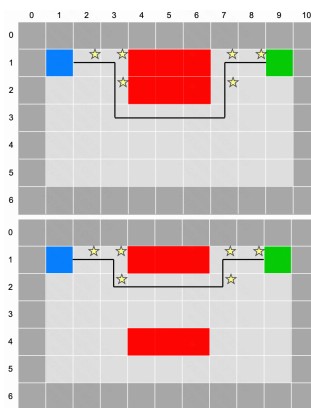

Figure 1: Lava world example with dynamics shift.

The state trajectories of the optimal policies on two example lava environments are plotted with black lines. The optimal state distributions are different under distinct environment dynamics. For

example, state (3,3,0) has non-zero probability under $d^*(s)$ in Fig. 1 (above), but cannot be visited by the optimal policy in Fig. 1 (bottom). Existing state-only policy transfer algorithms will therefore not be suitable for such seemingly simple task, which is also demonstrated by the empirical results in Sec. 5.1. The main cause of this distribution difference is the *break of accessibility*. State (4,2) is accessible from (3,2) in Fig. 1 (lower), but is inaccessible in Fig. 1 (upper). The inaccessible states will certainly have zero visitation probability and make the optimal state distribution different. Unfortunately, such break of accessibility can happen in various real-world tasks. We discuss more tasks with distinct optimal state distributions in Appendix B.

## 3.2 ANCHOR STATE ORIENTED POLICY REGULARIZATION

Motivated by examples in Sec. 3.1, we propose to ignore inaccessible states and focus on states that are accessible under all possible dynamics. We term the latter as *anchor states* with the following formal definition.

**Definition 3.1** (Anchor state). In an HiP-MDP $(\mathcal{S}, \mathcal{A}, \Theta, T, r, \gamma, \rho_0)$, state $s \in \mathcal{S}$ is called an *anchor state* if for all $\theta \in \Theta$, we have $d^*_{T_\theta}(s) > 0$. We denote $S^+ \subseteq S$ as the set of anchor states and $S^- := S - S^+$ as the set of non-anchor states. The *anchor state distribution* is defined as $d^{\pi,+}_T(s) = (1 - \gamma) \sum_{t=0}^{\infty} \gamma^t p\left(s_t = s, s_t \in S^+ | \pi, T\right) / Z(\pi)$, where $Z(\pi) = \sum_{t=0}^{\infty} \gamma^t p\left(s_t \in S^+ | \pi, T\right)$ is the normalizing term.

In the lava world example in Fig. 1, anchor states are marked with yellow stars. The stationary state distribution on non-anchor states $d^{\pi,-}_T(s)$ can be defined similarly. To learn from the anchor state distributions, we propose to regularize the training policy to generate a stationary state distribution that is close to the optimal anchor state distribution $d^{*,+}_T(s)$. With respect to the non-anchor states, the optimal stationary state distributions tend to be different across different dynamics. So instead of making policy regularization, we encourage the policy to generate new distributions on non-anchor states. The resulting constrained policy optimization problem is formulated as follows:

$$\max_{\pi} \, \mathbb{E}_{\theta, \tau_\pi} \left[ \sum_{t=0}^{\infty} \gamma^t r\left(s_t, a_t, s_{t+1}\right) \right] \quad \text{s.t.} \, \max_T D_{\mathrm{KL}}\left(d^\pi_T(\cdot) \| d^{*,+}_{T_0}(\cdot)\right) - D_{\mathrm{KL}}\left(d^\pi_T(\cdot) \| d^{*,-}_{T_0}(\cdot)\right) < \varepsilon,$$

(1)

where $T_0$ is an arbitrary environment dynamics. Eq. (1) can be transformed into an unconstrained optimization problem with the following Lagrangian:

$$L = -\mathbb{E}_{\theta, \tau_\pi, T} \left[ \sum_{t=0}^{\infty} \gamma^t \left( r(s_t, a_t, s_{t+1}) + \lambda \log \frac{d^{*,+}_{T_0}(s_t)}{d^\pi_T(s_t)} - \lambda \log \frac{d^{*,-}_{T_0}(s_t)}{d^\pi_T(s_t)} \right) \right] - \frac{\lambda \varepsilon}{1 - \gamma}, \quad (2)$$

where $\lambda > 0$ is the Lagrangian Multiplier. The only difference between Eq. (2) and standard RL's optimization objective is that the logarithms of state probability ratios are augmented to the environment reward $r(s_t, a_t, s_{t+1})$. Therefore, the proposed approach can be easily applied to a wide range of RL algorithms with reward augmentation, as demonstrated by the empirical results in Sec. 5.

## 3.3 ESTIMATING DENSITY RATIOS WITH STATE UNCERTAINTY AND VALUE FUNCTION

One remaining challenge in optimizing Eq. (2) is that the density ratios contain intractable stationary state distributions and cannot be directly computed. In the following subsection, we discuss approaches for computing the density ratio $d^{*,+}_{T_0}(s_t)/d^\pi_T(s_t)$, and $d^{*,-}_{T_0}(s_t)/d^\pi_T(s_t)$ can be similarly obtained. Motivated by recent advances in computing likelihood-free importance weights (Nguyen et al., 2010), we propose a data-based approach for estimating the density ratio. The following lemma indicates that the density ratio can be obtained through maximizing the discrepancy of two expectations that can be estimated by sampling from two datasets.

**Lemma 3.2** (Nguyen et al. (2010)). *Assume that function $f$ has first order derivatives $f'$ at $[0, +\infty)$. The f-divergence between two probabilistic measures $P, Q \in \mathcal{P}(\mathcal{S})$ is defined as $D_f(P\|Q) = \int_{\mathcal{S}} f(\mathrm{d}P(s)/\mathrm{d}Q(s))\mathrm{d}Q(s)$. Then for all $P, Q \in \mathcal{P}(\mathcal{S})$ and $\omega : \mathcal{S} \to \mathbb{R}^+$, $D_f(P\|Q) \geq \mathbb{E}_P\left[f'(\omega(s))\right] - \mathbb{E}_Q\left[f^*\left(f'(\omega(s))\right)\right] (*)$, where $f^*$ denotes the convex conjugate of $f$ and the equality is achieved if and only if $\omega = \mathrm{d}P/\mathrm{d}Q$.*

According to Lem. 3.2, if $P$ is the optimal anchor state distribution $d^{*,+}_{T_0}(s_t)$, $Q$ is the stationary state distribution of policy $d^\pi_T(s_t)$, and the R.H.S. of Eq.(*) is maximized, $\mathrm{d}P/\mathrm{d}Q$ is exactly the density

---

**Algorithm 1** The workflow of ASOR on top of ESCP (Luo et al., 2022).

---

1: **Input:** Training MDPs $\{\mathcal{M}_0, \cdots, \mathcal{M}_{n-1}\}$; Context encoder $\phi$; Policy network $\pi$; Value network $V$; Density Ratio Network $\omega^+, \omega^-$; Rollout horizon $H$; State partition ratio $\rho_1, \rho_2$; Regularization coefficient $\lambda$; Replay Buffer $\mathcal{R}$.
2: **for** $step = 0, 1, 2, \ldots$ **do**
3:    Sample MDP $\mathcal{M}_i$ from $\{\mathcal{M}_0, \mathcal{M}_1, \cdots, \mathcal{M}_{n-1}\}$ uniformly.
4:    **for** $t = 1, 2, \ldots, H$ **do**
5:       Sample $z_t$ from $\phi(z \mid s_t, a_{t-1}, z_{t-1})$ and then sample $a_t$ from $\pi(a \mid s_t, z_t)$, as in ESCP.
6:       Rollout and get transition data $(s_{t+1}, r_t, d_{t+1}, s_t, a_t, z_t)$ from $\mathcal{M}_i$; Add the data to the replay buffer $\mathcal{R}$.
7:    Sample a batch $D_{\text{batch}}$; Add $\rho_1 \rho_2 |D_{\text{batch}}|$ states with top $\rho_1$ portion of high values and $\rho_2$ portion of high proxy visitation counts to $D_P^+$; Add other states to $D_Q^+$. Add $\rho_1(1-\rho_2)|D_{\text{batch}}|$ states with high values and low visitation counts to $D_P^-$; Add other states to $D_Q^-$.
8:    Train $\omega^+$ and $\omega^-$ to optimize the R.H.S. of Eq.($*$) in Lem. 3.2 with $D_P^+, D_Q^+$, and $D_P^-, D_Q^-$, respectively.
9:    For one-step transition in $D_{\text{batch}}$, update $r_t$ with $r_t + \lambda \log \omega^+(s_t) - \lambda \log \omega^-(s_t)$.
10:   Use the updated $D_{\text{batch}}$ to update $\phi$, $\pi$, and $V$.

---

ratio we would like to obtain. Therefore, the density ratios can be estimated through optimizing the R.H.S. of Eq.(*) with respect to $\omega(s)$, which can be represented by a neural network. Such an optimization process requires access to expectations on $d_{T_0}^{*,+}(\cdot)$ and $d_T^\pi(\cdot)$ that can be approximated by learning with data that are likely to be sampled from them.

While $d_T^\pi(\cdot)$ can be related to data newly collected in the replay buffer (Liu et al., 2021; Sinha et al., 2022), sampling from $d_{T_0}^{*,+}(\cdot)$ is still challenging. We introduce the binary observation state $\mathcal{O}_t$ with $\mathcal{O}_t = 1$ denoting $s_t$ is the optimal state at timestep $t$ (Levine, 2018). $d_{T_0}^{*,+}(s_t)$ can therefore be written as $d_{T_0}^{\pi,+}(s|\mathcal{O}_{0:\infty})$, which is the distribution of the states generated by a certain $\pi$, given these states are optimal. With the Bayes' rule, we have

$$
\begin{aligned}
\frac{d_{T_0}^{*,+}(s_t)}{d_T^\pi(s)} &= \frac{d_{T_0}^{\pi,+}(s|\mathcal{O}_{0:\infty})}{d_T^\pi(s)} = \frac{p(\mathcal{O}_{0:\infty}|s, \pi, T_0) d_{T_0}^{\pi,+}(s)}{p(\mathcal{O}_{0:\infty}|\pi, T_0)} \cdot \frac{1}{d_T^\pi(s)} \\
&= \frac{p(\mathcal{O}_{0:\infty}|s, \pi, T_0) d_T^\pi(s)}{p(\mathcal{O}_{0:\infty}|\pi, T_0)} \cdot \frac{1}{d_T^\pi(s)} \cdot \frac{d_{T_0}^{\pi,+}(s)}{d_T^\pi(s)} = \frac{d_{T_0}^*(s_t)}{d_T^\pi(s)} \cdot \frac{d_{T_0}^{\pi,+}(s)}{d_T^\pi(s)},
\end{aligned}
\tag{3}
$$

where the last equation is also obtained with the Bayes' rule. According to Xue et al. (2023a), state $s$ will be more likely to be sampled from $d_{T_0}^*(\cdot)$ than from $d_T^\pi(\cdot)$ if it has a higher state value $V(s)$ than average. Meanwhile, $d_{T_0}^{\pi,+}(s)$ will be higher if $s$ falls in the set of anchor states $S^+$ and is most likely to be visited by all optimal policies under dynamics shift. Therefore, any of the pseudo-count approaches of visitation frequency can be used to measure whether $s$ is likely to be sampled from $d_{T_0}^{\pi,+}(s)$. In simulated environments with small observation space (Sec. 5.1, 5.2, 5.3), disagreement in next state predictions of ensembled environment models has shown to be a good proxy of visitation frequency (Yu et al., 2020). In large-scale real-world tasks (Sec. 5.4), next state predictions can be unreliable, so we adopt Random Network Distillation (RND) (Burda et al., 2019) and use the error of predicting a random mapping as the proxy visitation measure.

### 3.4 PRACTICAL ALGORITHM

Summarizing previous derivations, we obtain a practical reward augmentation algorithm termed as ASOR (**Anchor State Oriented Regularization**) for policy optimization under dynamics shift. We select the ESCP (Luo et al., 2022) algorithm, which is one of the SOTA algorithms in online cross-dynamics policy training, as the base algorithm. The detailed procedure of ESCP+ASOR is shown in Alg. 1. After the environment rollout and obtaining the replay buffer (line 6), we sample a batch of data from the buffer, obtain a portion of $\rho_1 \rho_2$ states with higher values and proxy visitation counts, and add them to the dataset $\mathcal{D}_P^+$. Other states are added to $\mathcal{D}_Q^+$. A portion of $\rho_1(1-\rho_2)$ states with higher values and lower proxy visitation counts are added to the dataset $\mathcal{D}_P^-$. Other states

are added to $\mathcal{D}_Q^-$ (line 7). We set $\rho_1 = \rho_2 = 0.5$ in offline experiments with medium-expert level of data. $\rho_1 = \rho_2 = 0.3$ is set in all other experiments. Then density ratio networks $\omega^+$ and $\omega^-$ are trained (line 8). While there are multiple choices of $f$ in Lem. 3.2 (Nowozin et al., 2016), we set it to $f(u) = u \log u - (u+1) \log(u+1)$ in accordance with GAN's setup (Goodfellow et al., 2014) for convenient implementation. Density ratio networks estimate the logarithm of the state density ratios $\lambda \log \omega^+(s) - \lambda \log \omega^-(s)$, which are added to the reward $r_t$ (line 9). $\lambda$ is regarded as a hyperparameter with values 0.1 or 0.3. The effect of $\rho_1$, $\rho_2$, and $\lambda$ is investigated in Sec. 5.2. The procedure of the offline algorithm MAPLE (Lee et al., 2020)+ASOR is similar to ESCP+ASOR, where the datasets $D_P^+, D_Q^+, D_P^-, D_Q^-$ are built with data from the offline dataset, instead of the replay buffer.

## 4 THEORETICAL ANALYSIS

In this section, we provide theoretical justifications for the policy regularization approach in Sec. 3. The notations are introduced in Sec. 2.1 and proofs can be found in Appendix A.2. Thm. 4.2 shows that regularized with the optimal anchor state distribution, the learning policy can obtain a stronger performance lower-bound than previous analysis (Xu et al., 2023; Yu et al., 2020). The theorem also features a weaker assumption on MDP accessibility than that in SRPO (Xue et al., 2023a). Moreover, Thm. 4.4 provides finite-sample analysis for the policy regularization. We start with the definition of $M$-$R_s$ accessible MDPs, which formally characterizes the "similar structure" required by MDPs with different dynamics to have closely related optimal state distribution.

**Definition 4.1.** Consider MDPs $\mathcal{M}_1 = (\mathcal{S}, \mathcal{A}, T_1, r, \gamma, \rho_0)$ and $\mathcal{M}_2 = (\mathcal{S}, \mathcal{A}, T_2, r, \gamma, \rho_0)$. If for all $k \in \mathbb{R}^+$, states $s_0, s_1, \cdots, s_k \in \mathcal{S}^+$, and actions $a_0, a_1, \cdots, a_{k-1} \in \mathcal{A}$ such that $\prod_{n=1}^{k} T_1(s_n | s_{n-1}, a_{n-1}) > 0$, there exists $N \in \mathbb{R}^+$, states $s_0', s_1', \cdots, s_{N+k-1}'$, and actions $a_0', a_1', \cdots, a_{N+k-2}'$ such that $N \leqslant M$, $s_0 = s_0'$, $s_k = s_{N+k-1}'$, $\prod_{n=1}^{N+k-1} T_2(s_n' | s_{n-1}', a_{n-1}') > 0$, and $\left| \sum_{n=1}^{N+k-2} \gamma^{n-1} r(s_n', a_n', s_{n+1}') - \sum_{n=1}^{k-1} \gamma^{n-1} r(s_n, a_n, s_{n+1}) + (1 - \gamma^{N-1}) V_{T_1}^*(s_0) \right| \leqslant R_s$, $\mathcal{M}_2$ is referred to as $M$-$R_s$ accessible from $\mathcal{M}_1$.

In this definition, $M$ is the number of extra steps required in $\mathcal{M}_2$ to reach the state $s_k$ from $s_0$, compared with in $\mathcal{M}_1$. $R_s$ constrains the reward discrepancy in these extra steps. One special case is when $\mathcal{M}_2$ and $\mathcal{M}_1$ are 1-0 accessible from each other, all states between $(s_0, s_k)$ will have the same state accessibility. It is identical to the property of "homomorphous MDPs" (Xue et al., 2023a), based on which a theorem about identical optimal state distribution can be proved. Most of the previous approaches in IfO is built upon such assumption of 1-0-accessible MDPs, which is an over-simplification in many tasks. For example, the Minigrid environment in Sec. 3.1 and Sec. 5.1 contains 3-0.03 accessible MDPs. Instead, the following theorem is based on the milder assumption on $M$-$R_s$ accessible MDPs, where we show that the learning policy $\hat{\pi}$ will have a performance lower-bound given a bounded KL-divergence with the optimal anchor state distribution.

**Theorem 4.2.** *Consider the MDP $\mathcal{M}_1 = (\mathcal{S}, \mathcal{A}, T_1, r, \gamma, \rho_0)$ which is $M$-$R_s$ accessible from the MDP $\mathcal{M}_2 = (\mathcal{S}, \mathcal{A}, T_2, r, \gamma, \rho_0)$. For all policy $\hat{\pi}$, if there exists one certain dynamics $T_0$ such that $\max_T D_{\mathrm{KL}}(d_T^{\hat{\pi}}(\cdot) \| d_{T_0}^{*,+}(\cdot)) \leqslant \varepsilon$, we have*

$$\eta(\hat{\pi}) \geqslant \max_T \eta(\pi_T^*) - \frac{2R_s + 6\lambda + \sqrt{2} R_{\max} \sqrt{\varepsilon}}{1 - \gamma}. \tag{4}$$

Previous approaches (Xu et al., 2023; Janner et al., 2019; Xue et al., 2023c) also provide policy performance lower-bounds, but these bounds have quadratic dependencies on the effective planning horizon $\frac{1}{1-\gamma}$. By anchor state-based policy regularization, we obtain a tighter discrepancy bound with linear dependency on the effective horizon.

The following theorem analyses the performance lower-bound of $\hat{\pi}$ if it is regularized with finite samples from the optimal anchor state distribution. Due to the poor generalization ability of KL distance (Arora et al., 2017; Xu et al., 2020), we characterize the regularization error in Eq. (1) with the network distance (Arora et al., 2017).

**Definition 4.3** (Neural network distance (Arora et al., 2017))**.** For a class of neural networks $\mathcal{P}$, the neural network distance between two state distributions, $\mu$ and $\nu$, is defined as

$$d_{\mathcal{P}}(\mu, \nu) = \sup_{P \in \mathcal{P}} \{\mathbb{E}_{s \sim \mu}[P(s)] - \mathbb{E}_{s \sim \nu}[P(s)]\}. \tag{5}$$

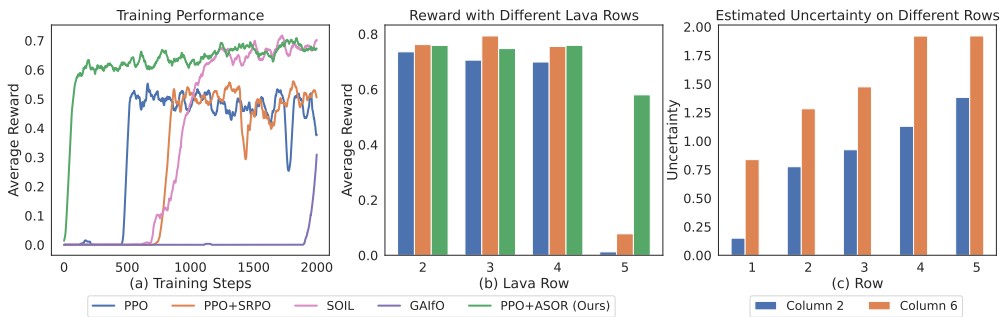

Figure 2: Results in the Minigrid environment. (a) Performance comparison between PPO+ASOR and baseline algorithms. (b) The average reward on the environment with different position of the second lava row. PPO and PPO+SRPO has very low rewards when the bottom lava is on row 5; (c) The state uncertainty estimated by ASOR on different rows of the lava environment.

For better characterization of the reward function with the network set $\mathcal{P}$, we introduce the linear span of $\mathcal{P}$ (Xu et al., 2020) as $\text{span}(\mathcal{P}) = \{c_0 + \sum_{i=1}^n c_i P_i : c_0, c_i \in \mathbb{R}, P_i \in \mathcal{P}, n \in \mathbb{N}\}$, leading to the following theorem.

**Theorem 4.4.** *Consider the MDP $\mathcal{M}_1 = (\mathcal{S}, \mathcal{A}, T_1, r, \gamma, \rho_0)$ which is $M\text{-}R_s$ accessible from the MDP $\mathcal{M}_2 = (\mathcal{S}, \mathcal{A}, T_2, r, \gamma, \rho_0)$ and the network set $\mathcal{P}$ bounded by $\Delta$, i.e., $|P(s)| \leqslant \Delta$. Given $\{s^{(i)}\}_{i=1}^m$ sampled from $d_{T_0}^{+,*}$, if the reward function $r_{\hat{\pi},T}(s) = \mathbb{E}_{a \sim \hat{\pi}, s' \sim T} r(s, a, s')$ lies in the linear span of $\mathcal{P}$, for policy $\hat{\pi}$ regularized by $\hat{d}_{T_0}^{+,*}$ with the constraint $\max_T d_{\mathcal{P}}(\hat{d}_T^{\hat{\pi}}, \hat{d}_{T_0}^{+,*}) < \varepsilon_{\mathcal{P}}$, we have*

$$\eta(\hat{\pi}) \geqslant \max_T \eta_T(\pi_T^*) - \frac{2R_s + 8\lambda}{1 - \gamma} - \frac{2\|r\|_{\mathcal{P}}}{1 - \gamma} \left( \hat{\mathcal{R}}_{d_{T_2}^{+,*}}^{(m)}(\mathcal{P}) + \hat{\mathcal{R}}_{d_{T_1}^{\hat{\pi}}}^{(m)}(\mathcal{P}) + 6\Delta \sqrt{\frac{\log(2/\delta)}{m}} + \frac{\varepsilon_{\mathcal{P}}}{2} \right) \quad (6)$$

*with probability at least $1 - \delta$, where $d_{\mathcal{P}}$ is the network distance (Arora et al., 2017), $\hat{d}_T^{\hat{\pi}}$ and $\hat{d}_{T_0}^{+,*}$ are the empirical version of distributions $d_T^{\hat{\pi}}$ and $d_{T_0}^{+,*}$ on $\{s^{(i)}\}_{i=1}^m$, $\hat{\mathcal{R}}$ is the empirical Rademacher complexity, and $\|r\|_{\mathcal{P}} = \inf \{\sum_{i=1}^n |c_i| : r = \sum_{i=1}^n c_i P_i + c_0, \forall n \in \mathbb{N}, c_0, c_i \in \mathbb{R}, P_i \in \mathcal{P}\}$.*

Thm. 4.4 shows that with finite samples, the anchor-based policy regularization still leads to a tight performance lower-bound with linear horizon dependency. The lower-bound is stronger than the sample complexity analysis of Behavior Cloning with quadratic horizon dependency (Xu et al., 2020) and has the same horizon dependency with GAIL (Ho & Ermon, 2016). Meanwhile, ASOR has a more stable regularization process than GAIL due to the non-adversarial way of generating $D_P$ and $D_Q$, as demonstrated in Fig. 4 (right).

## 5 EXPERIMENTS

In this section, we conduct experiments to investigate the following questions: (1) Can ASOR efficiently learn from data with dynamics shift and outperform current state-of-the-art algorithms? (2) Is ASOR general enough when applied to different styles of training environments, various sources of environment dynamics shift, and when combined with distinct algorithm setup? (3) How does each component of ASOR (e.g., the reward augmentation and the pseudo-count of state visitations) and its hyperparameters perform in practice? To answer questions (1)(2), we construct cross-dynamics training environments based on tasks including Minigrid (Chevalier-Boisvert et al., 2023), D4RL (Fu et al., 2020), MuJoCo (Todorov et al., 2012), and a Fall Guys-like Battle Royal Game. Dynamics shift in these environments comes from changes in navigation maps, evolvements of environment parameters, and different layouts of obstacles. To train RL policies in these environments, ASOR is implemented on top of algorithms including PPO (Schulman et al., 2017), MAPLE (Chen et al., 2021), and ESCP (Luo et al., 2022), which are all state-of-the-art approaches in the corresponding field. To answer question (3), we visualize how the density ratio estimator $\omega^+$, $\omega^-$ and the pseudo state count behave in different environments. Moreover, ablation studies are conducted to examine the role of the density ratio estimator and the influence of hyperparameters. Detailed descriptions of baseline algorithms are in Appendix C.1.

### 5.1 RESULTS IN MINIGRID ENVIRONMENT

For experiments in the Minigrid environment (Chevalier-Boisvert et al., 2023), the row number of the bottom lava is randomly sampled from $\{2, 3, 4, 5\}$, leading to dynamics shift. By including

Table 1: Results of offline experiments on MuJoCo tasks. Numbers before $\pm$ are scores normalized according to D4RL (Fu et al., 2020) and averaged across trials with four different seeds. Numbers after $\pm$ are normalized standard deviations. ME, M, MR and R correspond to the medium-expert, expert, medium-replay and random dataset, respectively.

| | BCO | SOIL | CQL | MOPO | MAPLE | MAPLE +DARA | MAPLE +SRPO | MAPLE +ASOR |
|---|---|---|---|---|---|---|---|---|
| Walker2d-ME | 0.25±0.04 | 0.14±0.08 | **0.63**±0.13 | 0.06±0.05 | 0.14±0.08 | 0.31±0.02 | 0.22±0.07 | 0.29±0.12 |
| Walker2d-M | 0.17±0.07 | 0.16±0.01 | **0.75**±0.02 | 0.15±0.22 | 0.41±0.19 | 0.46±0.10 | 0.32±0.17 | 0.49±0.04 |
| Walker2d-MR | 0.01±0.00 | 0.04±0.01 | 0.06±0.00 | -0.00±0.00 | 0.13±0.01 | 0.12±0.00 | 0.13±0.01 | **0.14**±0.01 |
| Walker2d-R | 0.00±0.00 | 0.00±0.00 | 0.00±0.00 | -0.00±0.00 | **0.22**±0.00 | 0.16±0.01 | 0.22±0.00 | 0.22±0.00 |
| Hopper-ME | 0.08±0.02 | 0.01±0.00 | 0.20±0.07 | 0.01±0.00 | 0.45±0.07 | 0.49±0.01 | 0.43±0.06 | **0.51**±0.06 |
| Hopper-M | 0.00±0.00 | 0.08±0.00 | 0.29±0.06 | 0.01±0.00 | 0.38±0.09 | 0.26±0.02 | 0.48±0.04 | **0.71**±0.14 |
| Hopper-MR | 0.00±0.00 | 0.00±0.00 | 0.08±0.00 | 0.01±0.01 | 0.55±0.17 | 0.75±0.10 | 0.73±0.16 | **0.76**±0.08 |
| Hopper-R | 0.00±0.00 | 0.00±0.00 | 0.10±0.00 | 0.01±0.00 | 0.12±0.00 | 0.12±0.00 | 0.25±0.08 | **0.32**±0.00 |
| HalfCheetah-ME | 0.43±0.00 | 0.00±0.00 | 0.03±0.04 | -0.03±0.00 | 0.53±0.07 | 0.39±0.00 | 0.58±0.04 | **0.61**±0.02 |
| HalfCheetah-M | 0.14±0.02 | 0.39±0.00 | 0.42±0.01 | 0.36±0.27 | 0.61±0.01 | **0.66**±0.03 | 0.62±0.01 | 0.62±0.01 |
| HalfCheetah-MR | 0.16±0.01 | 0.25±0.00 | 0.46±0.00 | -0.03±0.00 | 0.52±0.01 | 0.53±0.02 | 0.54±0.00 | **0.56**±0.01 |
| HalfCheetah-R | 0.14±0.01 | 0.35±0.01 | -0.01±0.01 | -0.03±0.00 | 0.20±0.02 | 0.19±0.01 | **0.22**±0.00 | 0.21±0.00 |
| Average | 0.11 | 0.11 | 0.25 | 0.04 | 0.36 | 0.37 | 0.40 | **0.45** |

Table 2: Results of ablation studies in Offline MuJoCo tasks. The scores are averaged on each environment with different expert levels.

| | Fixed $\lambda = 0.1$ | Fixed $\lambda = 0.3$ | Random partition | Fixed $\rho_1 = 0$ | Fixed $\rho_2 = 0$ | Fixed $\rho_1, \rho_2 = 0.5$ | Fixed $\rho_1, \rho_2 = 0.3$ | MAPLE +ASOR |
|---|---|---|---|---|---|---|---|---|
| Walker2d | 0.22 | 0.25 | 0.26 | 0.22 | 0.29 | **0.30** | 0.26 | 0.28 |
| Hopper | 0.31 | 0.54 | 0.30 | 0.47 | 0.38 | 0.46 | 0.54 | **0.58** |
| HalfCheetah | 0.48 | 0.49 | 0.47 | 0.49 | 0.50 | 0.49 | 0.50 | **0.50** |
| Average | 0.34 | 0.43 | 0.34 | 0.40 | 0.39 | 0.42 | 0.43 | **0.45** |

lava indicator as part of the state input, the policy is fully aware of environment dynamics changes and the need of context encoders (Luo et al., 2022; Lee et al., 2020) is excluded. The categorical action space includes moving towards four directions. The reward function for each environment step is -0.02 and reaching the green goal grid will lead to an additional reward of 1. The episode terminates when the red lava or the green goal grid is reached. For baseline algorithms we select online RL algorithms PPO (Schulman et al., 2017) and PPO+SRPO (Xue et al., 2023a), as well as IfO algorithms SOIL (Gangwani & Peng, 2020) and GAIfO (Torabi et al., 2018b).

We demonstrate the experiment results in Fig. 2 (a). Our ASOR algorithm can increase the performance of PPO by a large margin, while SRPO can only make little improvement. This is because the optimal state distribution in different lava world environments will not be the same. SRPO will still blindly consider all relevant states for policy regularization, leading to suboptimal policies. Fig. 2 (b) demonstrates the average reward with each possible position of the bottom lava block. PPO and PPO+SRPO have low performance when the bottom lava block is at Row 5. They mistakenly regard grids at (5,4) and (5,5) as optimal, but ASOR will recognize these grids as non-anchor states. We also demonstrate in Fig. 2 (c) the disagreement in next state predictions used to compute pseudo count. States far from the starting point have higher prediction disagreements and lower pseudo counts.

## 5.2 RESULTS IN OFFLINE RL BENCHMARKS

For offline RL benchmarks, we collect the static dataset from environments with three different environment dynamics in the format of D4RL (Fu et al., 2020). Specifically, data from the original MuJoCo environments, environments with 3 times larger body mass, and environments with 10 times higher medium density are included. For baseline algorithms, we inlude IfO algorithms BCO (Torabi et al., 2018a) and SOIL (Radosavovic et al., 2021), standard offline RL algorithms CQL (Kumar et al., 2020) and MOPO (Yu et al., 2020), offline cross-domain policy transfer algorithms MAPLE (Chen et al., 2021), MAPLE+DARA (Liu et al., 2022), and MAPLE+SRPO (Xue et al., 2023a).

The comparative results are exhibited in Tab. 1. IfO approaches have the worst performance because they ignore the reward information (BCO) or cannot safely exploit the offline dataset (SOIL). Without the ability of cross-domain policy learning, CQL and MOPO cannot learn from data with dynamics shift and show inferior performances. Cross-domain policy transfer algorithms MAPLE, MAPLE+DARA, and MAPLE+SRPO show reasonable performance enhancement, while

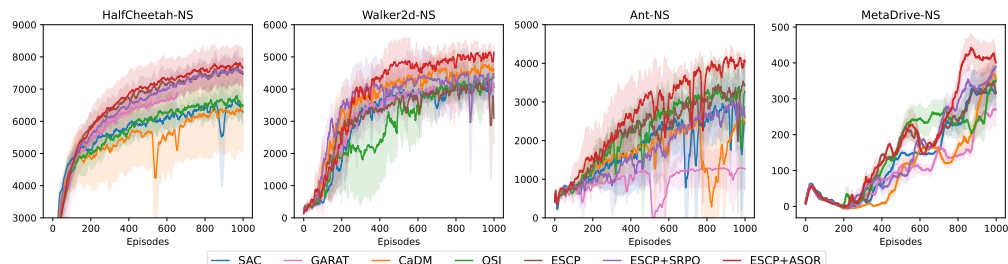

Figure 3: Results of online experiments on MuJoCo and MetaDrive tasks. "NS" refers to tasks with non-stationary environment dynamics.

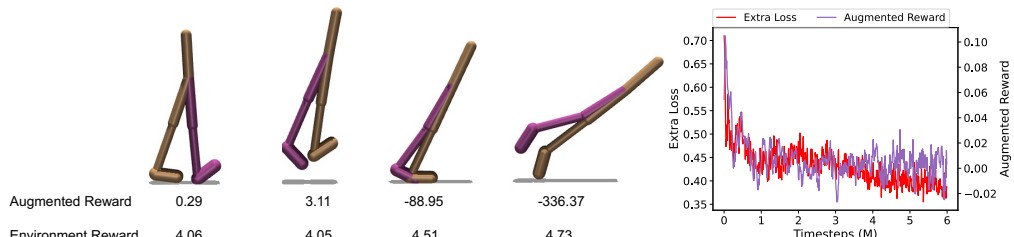

Figure 4: **Left**: Comparisons of the logarithm of density ratio, i.e., the augmented reward, and the environment reward on different states in the Walker-2d environment. The augmented reward can better reflect the state optimality. **Right**: Curves for average extra loss and augmented reward in the fall-guys like game environment.

our MAPLE+ASOR algorithm leads to the highest performance. This highlights the effectiveness of policy regularization on anchor states. We discuss the conceptual advantages of ASOR compared with DARA and SRPO in Appendix A.4.

The results of ablation studies are shown in Tab. 2. They show that a larger value of reward augmentation coefficient $\lambda$ can give rise to performance increase. Meanwhile, using fixed values of $\lambda$, $\rho_1$ and $\rho_2$ will not lead to a large drop of performance scores, so ASOR is robust to hyperparameter changes. ASOR's will have degraded performance if training datasets $D_P$ and $D_Q$ are improperly constructed, e.g., with random data partition, or without considering state values and visitation counts, i.e., with fixed $\rho_1 = 0$ or fixed $\rho_2 = 0$.

## 5.3 RESULTS IN ONLINE CONTINUOUS CONTROL TASKS

In online continuous control tasks, we explore other dimensions of dynamics shift, namely environment non-stationary and the continuous change of environment parameters. Such tasks are far more complicated than offline tasks with 3 different dynamics, but are within the capability of current approaches thanks to the existence of online interactive training environments. We consider the HalfCheetah, Walker2d, and Ant environments in the MuJoCo simulator (Todorov et al., 2012) and the autonomous driving environment in the MetaDrive simulator (Li et al., 2023). Sources of dynamics change include wind, joint damping, and traffic densities. For baselines we include the IfO algorithm GARAT (Desai et al., 2020), standard online RL algorithm SAC (Haarnoja et al., 2018), online cross-domain policy transfer algorithm OSI (Yu et al., 2017), ESCP (Luo et al., 2022), CaDM (Lee et al., 2020), and SRPO (Xue et al., 2023a).

Comparative results in online continuous control tasks are shown in Fig. 3, where our ESCP+ASOR algorithm has the best performance in all environments. Specifically, it only makes marginal improvements in the HalfCheetah environment, in contrast to large enhancement in other environments. This is because the agent will not "fall over" in the HalfCheetah environment, and the state accessibility will not change a lot under dynamics shift, undermining the effect of the anchor state-based policy regularization. We also compare in Fig. 4 (left) the augmented reward with the environment reward on different states in Walker-2d. On states where the agent is about to fall over, the augmented reward drops significantly while the environment reward does not change much, demonstrating the effectiveness of the reward augmentation.

## 5.4 RESULTS IN A LARGE-SCALE FALL GUYS-LIKE BATTLE ROYAL GAME ENVIRONMENT

Table 3: Experiment results in the fall guys-like game environment. Metrics with the up arrow (↑) are expected to have larger values and vice versa. Metrics with (∼) have no specific tendencies.

| | Total Reward (↑) | Goal Reward (↑) | Success Rate (↑) | Trapped Rate (↓) | Unnecessary Jump Rate (↓) | Distance from Cliff (∼) | Policy Entropy (∼) |
|---|---|---|---|---|---|---|---|
| PPO | 0.329±0.308 | 1.154±0.085 | 0.361±0.009 | 0.012±0.009 | 0.064±0.003 | 0.152±0.025 | 5.725±0.185 |
| PPO+SRPO | 0.337±0.257 | 1.513±0.076 | 0.376±0.006 | 0.038±0.015 | 0.040±0.003 | 0.148±0.018 | 5.859±0.098 |
| PPO+ASOR | **0.554**±0.336 | **1.781**±0.053 | **0.387**±0.005 | **0.005**±0.005 | **0.029**±0.003 | 0.143±0.021 | 6.358±0.122 |

In the large-scale fall guys-like game environment, we focus on highly dynamic and competitive race scenarios, characterized by a myriad of ever-changing obstacles, shifting floor layouts, and functional items. The elements within the game exhibit both functional and attribute changes, resulting in dynamics shift and evolving state accessibility. As shown in Fig. 5, the effects of trampolines

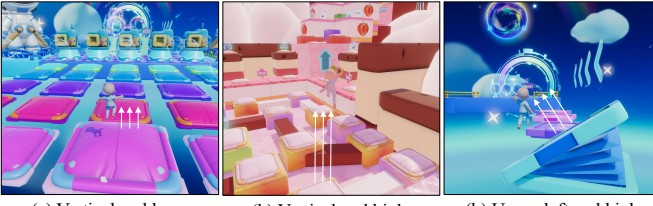

(a) Vertical and low   (b) Vertical and high   (b) Upper left and high

Figure 5: Demonstrations of dynamics shift caused by different trampoline effect. Colors and textures are only for visual enhancement and are not part of the agent's observations.

(e.g., height and orientation) vary across different maps and the agent's interaction with the trampoline will therefore result in different environment transitions depending on the specific configuration. The resulting dynamics shift has high stochasticity and cannot be effectively modelled by context encoder-based algorithms (Luo et al., 2022; Lee et al., 2020). We train the agent on 10 distinct maps, each presenting unique challenges and configurations. The training step is set to 6M. Metrics except policy entropy were averaged over the final 1M steps and the policy entropy is averaged in the initial 1M steps. More experiment details are listed in Appendix C.2, including additional map demonstrations, MDP setups, and the network structure.

As demonstrated in Tab. 3, PPO+ASOR achieves the highest scores in all five performance-related metrics. To be specific, the high total reward, unweighed goal reward, and success rate demonstrate the overall effectiveness of ASOR when applied to complex large-scale tasks. Low trapped rate, small distance from cliff, and high policy entropy demonstrate the strong exploration ability of ASOR since it is better at getting rid of low-reward regions and has higher policy stochasticity. The low unnecessary jump rate demonstrates the effectiveness of policy regularization only on anchor states. Jumping states may appear in the optimal trajectories in maps with diverse altitudes, but are unnecessary and hinder the fast goal reaching in other maps. Jumping states are regarded as non-anchor states in ASOR, where policy optimization will not be misled. Fig. 4 (right) shows the curve of the augmented reward and the extra loss, including the density ratio training loss and the RND training loss. The loss curve drops smoothly and the average augmented reward remains stable, which means that the density ratio estimation networks are easy to train and has stable performance.

## 6   CONCLUSION

In this paper, we focus on the problem of efficient policy optimization using data with dynamics shift. We demonstrate that existing IfO approaches are built upon the assumption of identical optimal state distribution, which can be unreliable because some states are no longer accessible when the environment dynamics changes. We remove this assumption and make policy regularization only on anchor states which can be reached by all optimal policies. By formally characterizing the difference of state accessibility under dynamics shift, we show that the anchor state-based regularization approach provides strong lower-bound performance guarantees for efficient policy optimization in the case of both perfect regularization and regularization on finite samples. In practice, the regularized policy optimization problem is transformed to the ASOR algorithm that can serve as an add-on reward augmentation module to existing RL approaches. Extensive experiments across various online and offline RL benchmarks o indicate ASOR can be effectively integrated with several state-of-the-art cross-domain policy transfer algorithms, substantially enhancing their performance.

**Limitations**   This paper focuses on the setting of HiP-MDP with evolving environment dynamics and a static reward function. The resulting ASOR algorithm will not be applicable to tasks with multiple reward functions. Meanwhile, the theoretical results will be weaker on some adversarial HiP-MDPs with large $R_s$. Details will be discussed in Appendix A.3.

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

# A    ADDITIONAL DERIVATIONS AND PROOFS

## A.1    DERIVATIONS OF THE LAGRANGIAN

For expression convenience, we denote $d_T^\pi(\cdot)$ with $d^\pi(\cdot)$, $d_{T_0}^{*,+}(\cdot)$ with $\delta^+$, and $d_{T_0}^{*,-}(\cdot)$ with $\delta^-$. We also omit the maximization over $T$ in Eq. (1) as it can be obtained by following all policy constrains in different dynamics. We start from the optimization problem

$$\max_\pi \ \mathbb{E}_{s_t, a_t, s_{t+1} \sim \tau_\pi} \left[ \sum_{t=0}^\infty \gamma^t r\left(s_t, a_t, s_{t+1}\right) \right] \quad \text{s.t.} \ \ D_{\mathrm{KL}}\left(d^\pi(\cdot) \| \delta^+(\cdot)\right) - D_{\mathrm{KL}}\left(d^\pi(\cdot) \| \delta^-(\cdot)\right) < \varepsilon. \tag{7}$$

The KL-Divergence term can be transformed as:

$$\begin{aligned} D_{\mathrm{KL}}\left(d^\pi(\cdot) \| \delta^+(\cdot)\right) &= -\mathbb{E}_{s \sim d^\pi(s)}\left[\log \delta^+(s) - \log d^\pi(s)\right] \\ &= -\int d^\pi(s)\left[\log \delta^+(s) - \log d^\pi(s)\right] ds \\ &= -\int (1-\gamma) \sum_{t=0}^\infty \gamma^t p\left(s_t = s\right)\left[\log \delta^+(s) - \log d^\pi(s)\right] ds \\ &= -(1-\gamma) \sum_{t=0}^\infty \int \gamma^t p\left(s_t = s\right)\left[\log \delta^+(s) - \log d^\pi(s)\right] ds \\ &= -(1-\gamma) \sum_{t=0}^\infty \mathbb{E}_{s_t \sim \tau_\pi}\left[\gamma^t \left(\log \delta^+(s_t) - \log d^\pi(s_t)\right)\right] \\ &= -(1-\gamma) \mathbb{E}_{s_t \sim \tau_\pi} \sum_{t=0}^\infty \gamma^t \left(\log \delta^+(s_t) - \log d^\pi(s_t)\right). \end{aligned} \tag{8}$$

So the constraint can be written as

$$-\mathbb{E}_{s_t \sim \tau_\pi} \sum_{t=0}^\infty \gamma^t \cdot \log \frac{\delta^+(s_t)}{d^\pi(s_t)} + \mathbb{E}_{s_t \sim \tau_\pi} \sum_{t=0}^\infty \gamma^t \cdot \log \frac{\delta^-(s_t)}{d^\pi(s_t)} - \frac{\varepsilon}{1-\gamma} < 0. \tag{9}$$

The optimization problem can be written as the following standard form

$$\begin{aligned} \min_\pi \ &\mathbb{E}_{s_t, a_t, s_{t+1} \sim \tau_\pi} \sum_{t=0}^\infty -\gamma^t r\left(s_t, a_t, s_{t+1}\right) \\ \text{s.t.} \ \ &-\mathbb{E}_{s_t \sim \tau_\pi} \sum_{t=0}^\infty \gamma^t \cdot \log \frac{\delta^+(s_t)}{d^\pi(s_t)} + \mathbb{E}_{s_t \sim \tau_\pi} \sum_{t=0}^\infty \gamma^t \cdot \log \frac{\delta^-(s_t)}{d^\pi(s_t)} - \frac{\varepsilon}{1-\gamma} < 0. \end{aligned} \tag{10}$$

So the Lagrangian $L$ is

$$L = -\mathbb{E}_{s_t, a_t, s_{t+1} \sim \tau_\pi} \left[ \sum_{t=0}^\infty \gamma^t \left( r(s_t, a_t, s_{t+1}) + \lambda \log \frac{\delta^+(s_t)}{d^\pi(s_t)} - \lambda \log \frac{\delta^-(s_t)}{d^\pi(s_t)} \right) \right] - \frac{\lambda \varepsilon}{1-\gamma}. \tag{11}$$

## A.2    PROOFS OF THEOREMS IN SEC. 4

**Lemma A.1** (Value Discrepancy). *Considering MDPs $\mathcal{M}_1 = (\mathcal{S}, \mathcal{A}, T_1, r, \gamma, \rho_0)$ and $\mathcal{M}_2 = (\mathcal{S}, \mathcal{A}, T_2, r, \gamma, \rho_0)$ which are $M$-$R_s$ accessible from each other, for all $s \in \mathcal{S}$ we have*

$$|V_{T_1}^*(s) - V_{T_2}^*(s)| \leqslant \frac{R_s + 2\lambda}{1-\gamma}, \tag{12}$$

*where $\lambda$ is the action coefficient in the reward function. Detailed definition are in Sec. 2.1.*

*Proof.* Without the loss of generality, we consider the state $s$ with $V_{T_1}^*(s) \geqslant V_{T_2}^*(s)$. Under the optimal policy $\pi_1^*(s)$, the next state of $s$ in $\mathcal{M}_1$ will be $s' = T(s, a^*)$. As $\mathcal{M}_2$ is $M$-$R_s$ accessible

from $\mathcal{M}_1$, there exists $N \leqslant M$ such that in $\mathcal{M}_2$, $s'$ can be reached from $s$ with action sequence $a_1, a_2, \cdots, a_N$. We then borrow the idea of iteratively computing $|V^*_{T_1}(s) - V^*_{T_2}(s)|$ from Xue et al. (Xue et al., 2023a). According to the optimistic Bellman equation

$$V^*_T(s) = \max_a \ r(s, a, s') + \gamma V^*_T(T(s, a)), \tag{13}$$

we have

$$
\begin{aligned}
&\left| V^*_{T_1}(s) - V^*_{T_2}(s) \right| \\
&= V^*_{T_1}(s) - V^*_{T_2}(s) \\
&\leqslant r(s, a^*, s') + \gamma V^*_{T_1}(s') - \sum_{i=0}^{N-1} \gamma^i r(s_i, a_i, s_{i+1}) - \gamma^N V^*_{T_2}(s') \\
&\quad (s_0 \doteq s, \ s_N \doteq s' \text{ for brevity}) \\
&= r(s, a^*, s') - \gamma^{N-1} r(s_{N-1}, a_{N-1}, s') + \gamma(1 - \gamma^{N-1}) V^*_{T_2}(s') \\
&\quad + \sum_{n=0}^{N-2} \gamma^n r(s_n, a_n, s_{n+1}) + \gamma[V^*_{T_1}(s') - V^*_{T_2}(s')] \\
&\leqslant (1 - \gamma^{N-1})(r(s, a^*, s') + \gamma V^*_{T_2}(s')) + 2\lambda + \sum_{n=0}^{N-2} \gamma^n r(s_n, a_n, s_{n+1}) + \gamma[V^*_{T_1}(s') - V^*_{T_2}(s')] \\
&\leqslant (1 - \gamma^{N-1})(r(s, a^*, s') + \gamma V^*_{T_1}(s')) + 2\lambda + \sum_{n=0}^{N-2} \gamma^n r(s_n, a_n, s_{n+1}) + \gamma[V^*_{T_1}(s') - V^*_{T_2}(s')] \\
&\leqslant R_s + 2\lambda + \gamma[V^*_{T_1}(s') - V^*_{T_2}(s')].
\end{aligned}
\tag{14}
$$

Iteratively computing $\left| V^*_{T_1}(s) - V^*_{T_2}(s) \right|$, we have

$$\left| V^*_{T_1}(s') - V^*_{T_2}(s') \right| \leqslant \frac{R_s + 2\lambda}{1 - \gamma}. \tag{15}$$

$\square$

**Theorem A.2** (Thm. 4.2 in the main paper.). *Consider the MDP $\mathcal{M}_1 = (\mathcal{S}, \mathcal{A}, T_1, r, \gamma, \rho_0)$ which is $M$-$R_s$ accessible from the MDP $\mathcal{M}_2 = (\mathcal{S}, \mathcal{A}, T_2, r, \gamma, \rho_0)$. For all policy $\hat{\pi}$, if there exists one certain dynamics $T_0$ such that $\max_T D_{\mathrm{KL}}(d^{\hat{\pi}}_T(\cdot) \| d^{*,+}_{T_0}(\cdot)) \leqslant \varepsilon$, we have*

$$\eta(\hat{\pi}) \geqslant \max_T \eta(\pi^*_T) - \frac{2R_s + 6\lambda + \sqrt{2} R_{\max} \sqrt{\varepsilon}}{1 - \gamma}. \tag{16}$$

*Proof.* $|\eta_{T_1}(\pi^*_{T_1}) - \eta_{T_2}(\pi^*_{T_2})|$ can be bounded with Thm. A.1:

$$|\eta_{T_1}(\pi^*_{T_1}) - \eta_{T_2}(\pi^*_{T_2})| = \left| \mathbb{E}_{s \in \rho_0} V^*_{T_1}(s) - \mathbb{E}_{s \in \rho_0} V^*_{T_2}(s) \right| \leqslant \frac{R_s + 2\lambda}{1 - \gamma}. \tag{17}$$

With a slight abuse of notation, we define the transition distribution $d^\pi_T(s, a, s') = d^\pi_T(s)\pi(a|s)T(s'|s, a)$ and the anchor-state transition distribution $d^{\pi,+}_{T_2}(s, a, s') = d^{\pi,+}_T(s)\hat{\pi}(a|s)T(s'|s, a)$. Consider $\tilde{\pi}$ such that $d^{\tilde{\pi}}_T(s) = d^{*,+}_{T_0}(s)$ for all $s \in \mathcal{S}$. The accumulated return of policy $\tilde{\pi}$ under transition $T_1$ can be written as $\eta_{T_1}(\hat{\pi}) = (1 - \gamma)^{-1} \mathbb{E}_{s, a, s' \sim d^{\tilde{\pi}}_{T_1}} [r(s, a, s')]$. We also consider the accumulated return of the optimal policy under transition $T_2$ including only anchor states: $\eta^+_{T_2}(\pi^{*,+}_{T_2}) = (1 - \gamma)^{-1} \mathbb{E}_{s, a, s' \sim d^{*,+}_{T_2}} [r(s, a, s')]$, where $\pi^{*,+}_{T_2}$ is the optimal policy making transitions among anchor states. Consider the Lipschitz property of the reward function:

$$|r(s, a_1, s') - r(s, a_2, s')| \leqslant \lambda \|a_1 - a_2\|_1. \tag{18}$$

Taking expectation w.r.t. $d^{\tilde{\pi}}_{T_1}(\cdot)$ on both sides, we get

$$\mathbb{E}_{s \sim d^{\tilde{\pi}}_{T_1}} |r(s, a_1, s') - r(s, a_2, s')| \leqslant \mathbb{E}_{s \sim d^{\tilde{\pi}}_{T_1}} \lambda \|a_1 - a_2\|_1. \tag{19}$$

Letting $\mu(A_1, A_2|s)$ be any joint distribution with marginals $\hat{\pi}$ and $\pi_{T_2}^{*,+}$ conditioned on $s \in S^+$. Taking expectation w.r.t. $\mu$ on both sides, we get

$$\left|\mathbb{E}_{d_{T_1}^{\hat{\pi}}} r(s, a, s') - \mathbb{E}_{d_{T_2}^{*,+}} r(s, a, s')\right| \leqslant \mathbb{E}_{s \sim d_{T'}^*} \mathbb{E}_{a_1, a_2 \sim \mu} |r(s, a_1, s') - r(s, a_2, s')|$$
$$\leqslant \lambda \mathbb{E}_{s \sim d_{T_1}^{\hat{\pi}}} E_\mu \|a_1 - a_2\|_1$$
$$\leqslant \max_s \lambda E_\mu \|a_1 - a_2\|_1 \tag{20}$$
$$\leqslant 2\lambda$$

According to the definitions of $\eta_{T_1}(\tilde{\pi})$ and $\eta_{T_2}^+(\pi_{T_2}^{+,*})$, the L.H.S. of Eq. (20) is exactly the difference of the two accumulated returns. Therefore, we get

$$\left|\eta_{T_1}(\tilde{\pi}) - \eta_{T_2}^+(\pi_{T_2}^{*,+})\right| \leqslant \frac{2\lambda}{1 - \gamma}. \tag{21}$$

Then we will compute the discrepancy between $\eta_{T_2}^+$ and $\eta_{T_2}$. $\eta_{T_2}$ can be computed with

$$\eta_{T_2}(\pi_{T_2}^*) = \mathbb{E}_{s \sim \rho_0} V_{T_2}^{\pi_{T_2}^*}(s)$$
$$= \mathbb{E}_\tau \sum_{n=0}^{N-1} \gamma^n r(s_n, a_n, s_{n+1}) + \gamma^N V_{T_2}^{\pi_{T_2}^*}(s_N), \tag{22}$$

where $s_N$ is the anchor state accessible from $s_0$ with $\pi_{T_2}^{*,+}$. According to the definition of $M$-$R_s$ accessible MDPs,

$$\eta_{T_2}(\pi_{T_2}^*) = \mathbb{E}_\tau \sum_{n=0}^{N-1} \gamma^n r_n + \gamma^N V_{T_2}^{\pi_{T_2}^*}(s_N) - R_s + R_s$$
$$\leqslant \mathbb{E}_\tau \sum_{n=0}^{N-1} \gamma^n r_n - \sum_{n=0}^{N-2} \gamma^n r_n - (\gamma^{N-1} - 1) r(s_0, \pi_{T_2}^{+,*}(s_0), s_N)$$
$$+ \gamma^N V_{T_2}^{\pi_{T_2}^*}(s_N) - (\gamma^N - \gamma) V_{T_2}^{\pi_{T_2}^{+,*}}(s_N) + R_s$$
$$\leqslant \mathbb{E}_\tau r(s_0, \pi_{T_2}^{+,*}(s_0), s_N) + \gamma V_{T_2}^{\pi_{T_2}^{+,*}}(s_N) + \gamma^N (V_{T_2}^{\pi_{T_2}^*}(s_N) - V_{T_2}^{\pi_{T_2}^{+,*}}(s_N)) + R_s + 2\lambda$$
$$= \eta_{T_2}^+(\pi_{T_2}^{+,*}) + \gamma^N (V_{T_2}^{\pi_{T_2}^*}(s_N) - V_{T_2}^{\pi_{T_2}^{+,*}}(s_N)) + R_s + 2\lambda, \tag{23}$$

where $r_n$ is the short for $r(s_n, a_n, s_{n+1})$. Iteratively scaling the value discrepancy between $\pi_{T_2}^*$ and $\pi_{T_2}^{+,*}$, we get

$$\left|\eta_{T_2}(\pi_{T_2}^*) - \eta_{T_2}^+(\pi_{T_2}^{+,*})\right| \leqslant \frac{R_s + 2\lambda}{1 - \gamma^M} \leqslant \frac{R_s + 2\lambda}{1 - \gamma}. \tag{24}$$

According to results in imitation learning (Lem. 6 in Xu et al. (2020)), we have

$$|\eta_{T_1}(\tilde{\pi}) - \eta_{T_1}(\hat{\pi})| \leqslant \frac{\sqrt{2} R_{\max} \sqrt{\varepsilon}}{1 - \gamma} \tag{25}$$

Combining Eq. (17)(21)(23)(25), we have

$$\left|\eta_{T_1}(\tilde{\pi}) - \eta_{T_1}(\pi_{T_1}^*)\right| \leqslant |\eta_{T_1}(\hat{\pi}) - \eta_{T_1}(\tilde{\pi})| + \left|\eta_{T_1}(\tilde{\pi}) - \eta_{T_2}^+(\pi_{T_2}^{+,*})\right|$$
$$+ \left|\eta_{T_2}^+(\pi_{T_2}^{+,*}) - \eta_{T_2}(\pi_{T_2}^*)\right| + |\eta_{T_2}(\pi_{T_2}^*) - \eta_{T_1}(\pi_{T_1}^*)| \tag{26}$$
$$\leqslant \frac{2R_s + 6\lambda + \sqrt{2} R_{\max} \sqrt{\varepsilon}}{1 - \gamma}.$$

Taking expectation with respect to all $T$ in the HiP-MDP concludes the proof. $\qquad \square$

**Lemma A.3** (Lemma 2 in Xu et. al (Xu et al., 2020)). *Consider a network class set $\mathcal{P}$ with $\Delta$-bounded value functions, i.e., $|P(s)| \leq \Delta$, for all $s \in \mathcal{S}, P \in \mathcal{P}$. Given an expert policy $\pi_E$ and*

an imitated policy $\pi_I$ with $d_{\mathcal{P}}\left(\hat{d}^{\pi_E}, \hat{d}^{\pi_1}\right) - \inf_{\pi \in \Pi} d_{\mathcal{P}}\left(\hat{d}^{\pi_E}, \hat{d}^{\pi}\right) \leq \varepsilon_{\mathcal{P}}$, then $\forall \delta \in (0, 1)$, with probability at least $1 - \delta$, we have

$$d_{\mathcal{P}}\left(d^{\pi_E}, d^{\pi_I}\right) \leq \inf_{\pi \in \Pi} d_{\mathcal{P}}\left(\hat{d}^{\pi_E}, \hat{d}^{\pi}\right) + 2\hat{\mathcal{R}}_{d^{\pi_E}}^{(m)}(\mathcal{P}) + 2\hat{\mathcal{R}}_{d^{\pi_I}}^{(m)}(\mathcal{P}) + 12\Delta\sqrt{\frac{\log(2/\delta)}{m}} + \varepsilon_{\mathcal{P}}. \quad (27)$$

*Proof.* See Appendix B.3 of Xu et al. (2020). $\square$

**Theorem A.4.** *Consider the MDP $\mathcal{M}_1 = (\mathcal{S}, \mathcal{A}, T_1, r, \gamma, \rho_0)$ which is $M$-$R_s$ accessible from the MDP $\mathcal{M}_2 = (\mathcal{S}, \mathcal{A}, T_2, r, \gamma, \rho_0)$ and the network set $\mathcal{P}$ bounded by $\Delta$, i.e., $|P(s)| \leqslant \Delta$. Given $\{s^{(i)}\}_{i=1}^m$ sampled from $d_{T_2}^{+,*}$, if $\pi_{T_2}^{+,*} \in \mathcal{P}$ and the reward function $r_{\hat{\pi}, T_1}(s) = \mathbb{E}_{a \sim \hat{\pi}, s' \sim T_1} r(s, a, s')$ lies in the linear span of $\mathcal{P}$, for policy $\hat{\pi}$ regularized by $\hat{d}_{T_2}^{+,*}$ according to Eq. (1) with $d_{\mathcal{P}}(\hat{d}_{T_1}^{\hat{\pi}}, \hat{d}_{T_2}^{+,*}) < \varepsilon_{\mathcal{P}}$, we have*

$$\eta_{T_1}(\hat{\pi}) \geqslant \eta_{T_1}(\pi_{T_1}^*) - \frac{2R_s + 8\lambda}{1 - \gamma} - \frac{2\|r\|_{\mathcal{P}}}{1 - \gamma}\left(\hat{\mathcal{R}}_{d_{T_2}^{+,*}}^{(m)}(\mathcal{P}) + \hat{\mathcal{R}}_{d_{T_1}^{\hat{\pi}}}^{(m)}(\mathcal{P}) + 6\Delta\sqrt{\frac{\log(2/\delta)}{m}} + \frac{\varepsilon}{2}\right) \quad (28)$$

*with probability at least $1 - \delta$.*

*Proof.* As $\mathcal{M}_1$ is $M$-$R_s$ accessible accessible from $\mathcal{M}_2$, there exists policy $\tilde{\pi}$ such that $d_{T_1}^{\tilde{\pi}}(s) = d_{T_2}^{*,+}(s)$ for all $s \in \mathcal{S}^+$. With Thm. A.2, we have

$$\eta_{T_1}(\tilde{\pi}) \geqslant \eta_{T_1}(\pi_{T_1}^*) - \frac{2R_s + 6\lambda}{1 - \gamma} \quad (29)$$

Then we compute the performance discrepancy $\eta_{T_1}(\hat{\pi}) - \eta_{T_1}(\tilde{\pi})$ given that $d_{\mathcal{P}}(\hat{d}_{T_1}^{\hat{\pi}}, \hat{d}_{T_1}^{\tilde{\pi}}) < \varepsilon_{\mathcal{P}}$. The following derivations borrow the main idea from Xu et al. (Xu et al., 2020) and turn the state-action occupancy measure $\rho$ into the state-only occupancy measure $d$. We start with the network distance of the ground truth state occupancy measures. According to Lem. A.3, we have

$$d_{\mathcal{P}}(d_{T_1}^{\hat{\pi}}, d_{T_1}^{\tilde{\pi}}) \leqslant 2\hat{\mathcal{R}}_{d_{T_2}^{+,*}}^{(m)}(\mathcal{P}) + 2\hat{\mathcal{R}}_{d_{T_1}^{\hat{\pi}}}^{(m)}(\mathcal{P}) + 12\Delta\sqrt{\frac{\log(2/\delta)}{m}} + \varepsilon_{\mathcal{P}} \quad (30)$$

with probability at least $1 - \delta$. Meanwhile,

$$|\eta_{T_1}(\hat{\pi}) - \eta_{T_1}(\tilde{\pi})|$$

$$\leqslant \frac{1}{1 - \gamma}\left|\mathbb{E}_{s \sim d_{T_1}^{\hat{\pi}}}\left[r_{\hat{\pi}, T_1}(s)\right] - \mathbb{E}_{s \sim d_{T_1}^{\tilde{\pi}}}\left[r_{\tilde{\pi}, T_1}(s)\right]\right|$$

$$\leqslant \frac{1}{1 - \gamma}\left|\mathbb{E}_{s \sim d_{T_1}^{\hat{\pi}}}\left[r_{\hat{\pi}, T_1}(s)\right] - \mathbb{E}_{s \sim d_{T_1}^{\tilde{\pi}}}\left[r_{\hat{\pi}, T_1}(s)\right]\right| + \frac{1}{1 - \gamma}\left|\mathbb{E}_{s \sim d_{T_1}^{\tilde{\pi}}}\left[r_{\hat{\pi}, T_1}(s)\right] - \mathbb{E}_{s \sim d_{T_1}^{\tilde{\pi}}}\left[r_{\tilde{\pi}, T_1}(s)\right]\right|$$

$$\leqslant \frac{1}{1 - \gamma}\left|\mathbb{E}_{s \sim d_{T_1}^{\hat{\pi}}}\left[r_{\hat{\pi}, T_1}(s)\right] - \mathbb{E}_{s \sim d_{T_1}^{\tilde{\pi}}}\left[r_{\hat{\pi}, T_1}(s)\right]\right| + \frac{2\lambda}{1 - \gamma}. \quad (31)$$

As we assume that the reward function $r_{\hat{\pi}, T_1}(s)$ lies in the linear span of $\mathcal{P}$, there exists $n \in \mathbb{N}, \{c_i \in \mathbb{R}\}_{i=1}^n$ and $\{P_i \in \mathcal{P}\}_{i=1}^n$, such that $r = c_0 + \sum_{i=1}^n c_i P_i$. So we obtain that

$$|\eta_{T_1}(\hat{\pi}) - \eta_{T_1}(\tilde{\pi})| \leqslant \frac{1}{1 - \gamma}\left|\mathbb{E}_{s \sim d_{T_1}^{\hat{\pi}}}\left[r_{\hat{\pi}, T_1}(s)\right] - \mathbb{E}_{s \sim d_{T_1}^{\tilde{\pi}}}\left[r_{\hat{\pi}, T_1}(s)\right]\right| + \frac{2\lambda}{1 - \gamma}$$

$$\leqslant \frac{1}{1 - \gamma}\left|\sum_{i=1}^n c_i \mathbb{E}_{s \sim d_{T_1}^{\hat{\pi}}}\left[P_i(s, a)\right] - \sum_{i=1}^n c_i \mathbb{E}_{s \sim d_{T_1}^{\tilde{\pi}}}\left[P_i(s, a)\right]\right| + \frac{2\lambda}{1 - \gamma}$$

$$\leqslant \frac{1}{1 - \gamma}\sum_{i=1}^n |c_i|\left|\mathbb{E}_{s \sim d_{T_1}^{\hat{\pi}}}\left[P_i(s, a)\right] - \mathbb{E}_{s \sim d_{T_1}^{\tilde{\pi}}}\left[P_i(s, a)\right]\right| + \frac{2\lambda}{1 - \gamma} \quad (32)$$

$$\leqslant \frac{1}{1 - \gamma}\left(\sum_{i=1}^n |c_i|\right) d_{\mathcal{P}}\left(d_{T_1}^{\hat{\pi}}, d_{T_1}^{\tilde{\pi}}\right) + \frac{2\lambda}{1 - \gamma}$$

$$\leqslant \frac{1}{1 - \gamma}\|r\|_{\mathcal{P}} d_{\mathcal{P}}\left(d_{T_1}^{\hat{\pi}}, d_{T_1}^{\tilde{\pi}}\right) + \frac{2\lambda}{1 - \gamma}.$$

Table 4: Comparison between the Lipschitz coefficient $\lambda$ and the maximum reward $R_{\max}$ in practical environments.

| Environment | Action-related Reward | $\lambda$ | $R_{\max}$ |
|---|---|---|---|
| CartPole-v0 | 0 | 0 | 1.00 |
| InvertedPendulum-v2 | 0 | 0 | 1.00 |
| Lava World | 0 | 0 | 1.00 |
| MetaDrive | 0 | 0 | $\geqslant 1$ |
| Fall-guys Like Game | 0 | 0 | $\geqslant 1$ |
| Swimmer-v2 | $-0.0001\|a\|_2^2$ | 0.0001 | 0.36 |
| HalfCheetah-v2 | $-0.1\|a\|_2^2$ | 0.1 | 4.80 |
| Hopper-v2 | $-0.001\|a\|_2^2$ | 0.001 | 3.80 |
| Walker2d-v2 | $-0.001\|a\|_2^2$ | 0.001 | $\geqslant 4$ |
| Ant-v2 | $-0.5\|a\|_2^2$ | 0.5 | 6.00 |
| Humanoid-v2 | $-0.1\|a\|_2^2$ | 0.1 | $\geqslant 8$ |

Combining Eq. (30) and Eq. (32), we have

$$\eta_{T_1}(\hat{\pi}) \geqslant \eta_{T_1}(\tilde{\pi}) - \frac{2\|r\|_{\mathcal{P}}}{1-\gamma}\left(\hat{\mathcal{R}}_{d_{T_2}^{+,*}}^{(m)}(\mathcal{P}) + \hat{\mathcal{R}}_{d_{T_1}^{\hat{\pi}}}^{(m)}(\mathcal{P}) + 6\Delta\sqrt{\frac{\log(2/\delta)}{m}} + \frac{\varepsilon}{2}\right) + \frac{2\lambda}{1-\gamma} \quad (33)$$

with probability at least $1 - \delta$. Combining Eq. (33) and Eq. (29), we have

$$\eta_{T_1}(\hat{\pi}) \geqslant \eta_{T_1}(\pi_{T_1}^*) - \frac{2R_s + 8\lambda}{1-\gamma} - \frac{2\|r\|_{\mathcal{P}}}{1-\gamma}\left(\hat{\mathcal{R}}_{d_{T_2}^{+,*}}^{(m)}(\mathcal{P}) + \hat{\mathcal{R}}_{d_{T_1}^{\hat{\pi}}}^{(m)}(\mathcal{P}) + 6\Delta\sqrt{\frac{\log(2/\delta)}{m}} + \frac{\varepsilon}{2}\right) \quad (34)$$

with probability at least $1 - \delta$. Taking expectation with respect to all $T$ in the HiP-MDP concludes the proof. $\qquad\square$

### A.3 Discussions on the Theorems

**Lipschitz Assumption** The Lipschitz assumption in Sec. 2.1 requires that if $s$ and $s'$ keep unchanged, the deviation of the reward $r$ will not be larger than $\lambda$ times the deviation of the action $a$:

$$|r(s, a_1, s') - r(s, a_2, s')| \leqslant \lambda\|a_1 - a_2\|_1. \quad (35)$$

Therefore, the Lipschitz coefficient $\lambda$ is only depends action-related terms in the reward function. In Tab. 4, we list the action-related terms of the reward functions for various RL evaluation environments, along with the corresponding values of $\lambda$ derived from these terms. As indicated in the table, the action-related terms in reward functions exhibit reasonably small coefficients in all environments compared with the maximum environment reward $R_{\max}$. Therefore, the Lipschitz coefficient $\lambda$ will not dominate the error term in Thm. 4.2 and Thm. 4.4.

**Failure Cases** Apart from the Lipschitz assumptions that can easily be realized, Thm. 4.2 and Thm. 4.4 depend on the formulation of $M$-$R_s$ accessible MDPs. Potential failure cases will therefore include tasks with high $R_s$, so that the performance lower bounds become weak. This will happen if states with lowest rewards exist in the optimal trajectory of some, and not all dynamics. In the example of lava world in Fig. 1, if a reward of -100 is assigned to grid (3,4), $R_s$ will be as large as 100, leading to a weak performance lower bound when the bottom lava block is at Row 2 with a best episode return of 1. Nevertheless, issues with theoretical analyses will not negatively influence the practical performance of ASOR.

### A.4 Comparisons with Previous Approaches

Intuitively, the practical algorithm procedure of ASOR share some insights with some offline RL algorithms including AWAC (Nachum et al., 2019), CQL (Kumar et al., 2020) and MOPO (Yu et al., 2020). For example, ASOR prefers states with high values similar to AWAC and states with high

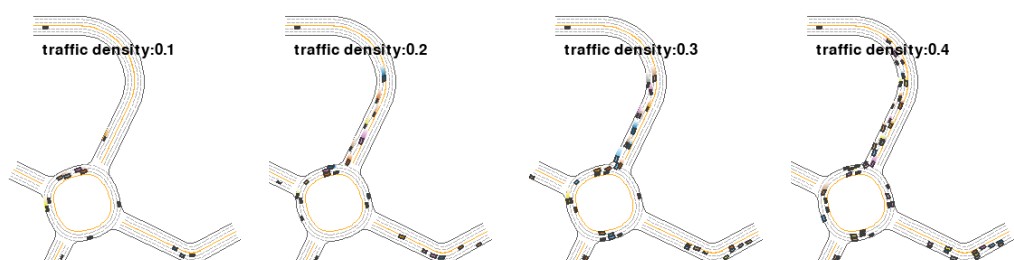

Figure 6: MetaDrive environments with different traffic densities.

visitation counts similar to CQL and MOPO. The advances of ASOR include: 1) By restricting the considered states to anchor states, ASOR can be applied in offline datasets collected under dynamics shift, where the aforementioned offline RL algorithms can only learn from the dataset with static dynamics. Thm. 4.2 and Thm. 4.4 demonstrate the effectiveness of such procedure. 2) ASOR modifies the original policy optimization process by reward augmentation. This enables the easy combination of ASOR with other cross-domain algorithms to enhance their performance.

ASOR also share the approach of classifier-based reward augmentation with DARA (Liu et al., 2022) and SRPO (Xue et al., 2023a). The classifier input in DARA is $(s, a, s')$ from the source and target environments. Compared with ASOR, the classifier in DARA exhibits higher complexity and is harder to train. It also requires the access to the information of target environments. Therefore, DARA has poor performance as demonstrated in Sec. 5.2 and cannot be applied to tasks with no prior knowledge on the target environments. The algorithm and theories of SRPO are based on the assumption of the same state accessibility, which is an over-simplification of some environments, as demonstrated in Sec. 3.1 and Sec. B. Comparative results in Sec. 5 demonstrate the inferior performance of SRPO compared with ASOR, in correspondence with the flaw in the assumption. Also, the theoretical analysis in the SRPO paper is built on the assumption called "homomorphous MDPs" which is stronger than the $M\text{-}R_s$ accessible MDPs used in this paper and is a special case of the latter.

## B    EXAMPLES OF DISTINCT STATE DISTRIBUTIONS

We claim in the main paper that previous assumption of similar state distributions under distribution shift will not hold in many scenarios. Apart from the motivating example of lava world in Sec. 3.1, we demonstrate more examples in the MetaDrive (Li et al., 2023) and the fall-guys like game environment. Examples of MetaDrive environments with different traffic densities are shown in Fig. 6. The dynamics shift lies in that the ego vehicle will have different probabilities to detect other vehicles nearby. In environments with low traffic densities, there is enough space for some vehicles with optimal policies to drive in high speeds. But in environments packed with surrounding vehicles, fast driving will surely lead to collisions. So the vehicles can only drive in low speeds. As the vehicle speed is included in the agent's state space, difference in traffic densities will lead to distinct optimal state distributions.

Visualizations of the fall-guys like game environment used in Sec. 5.4 are shown in Fig. 7, where map components including the conveyor belt speed, the balloon reaction, the floor reaction, and the hammer distance will work together, giving rise to dynamics shift. Taking the variation of hammer distance (Fig. 7 (d)) as an example, in the left environment the optimal trajectory will contain states where the hammer is near the agent. But in the right environment, there are trajectories that keep the hammer far away to avoid being hit out of the playground. Blindly imitating optimal states collected in the left environment will lead to suboptimal performance in the right environment.

## C  Experiment Details

### C.1  Baseline Algorithms

In experiments with four different tasks, we compare ASOR with the following baseline algorithms:

- PPO (Schulman et al., 2017): The widely used, off-the-shelf online RL algorithm with on-policy policy update.
- SAC (Haarnoja et al., 2018): The widely used off-policy RL algorithm with entropy maximization for better exploration.
- BCO (Torabi et al., 2018a): Learn a agent-specific inverse dynamics model to infer the experts' missing action information.
- GAIfO (Torabi et al., 2018b): A state-only version of the GAIL algorithm.
- GARAT (Desai et al., 2020): Use the action transformer trained with GAIL-like imitation learning to recover the experts' next states in the original environment.
- SOIL (Gangwani & Peng, 2020): An algorithm combining state-only imitation learning with policy gradients. The overall gradient consists of a policy gradient term and an auxiliary imitation term.
- CQL (Kumar et al., 2020): The widely used offline RL algorithm with conservative Q-learning.
- MOPO (Yu et al., 2020): A model-based offline RL algorithm subtracting disagreements in next-state prediction from environment rewards.
- MAPLE (Chen et al., 2021): The offline RL algorithm based on MOPO with an additional context encoder module for cross-dynamics policy adaptation.
- OSI (Yu et al., 2017): An algorithm using context encoders for online system identification.
- CaDM (Lee et al., 2020): The online RL algorithm with context encoders for cross-dynamics policy adaptation.
- DARA (Liu et al., 2022): Make reward augmentations with importance weights between source and target dynamics.
- SRPO (Xue et al., 2023a): Make reward augmentations with the assumption of similar optimal state distributions under dynamics shift.

### C.2  Additional Setup of the Fall-guys Like Game Environment

**Additional Environment Demonstrations**  Next, we present additional examples of dynamic shifts within the fall-guys like game environment to demonstrate the diverse and variable nature of in-game elements. As shown in Fig. 7, the game environment features a range of dynamic shifts which contribute to the complexity and unpredictability of the gameplay. Specifically, we observe the following scenarios: **Fig. 7 (a)**: The speed of conveyor belts changes across different game settings, leading to varied transitions in the agent's position and momentum when it steps onto these belts. **Fig. 7 (b)**: Balloons exhibit different reactions upon interaction with the agent. This variation can significantly affect the agent's subsequent trajectory. **Fig. 7 (c)**: The behavior of floors under the agent's influence varies significantly. Some floors may collapse, disappear, or shift unexpectedly, introducing further complexity to the environment. **Fig. 7 (d)**: The distance and direction in which the agent is ejected when struck by hammers can vary widely. This variability depends on the unpredictable environmental dynamic shifts, for example, the force and angle of the hammer's swing.

**MDP Setup**  Below, we provide definitions of state space, action space, and rewards in the fall-guys like game environment.

**State space $\mathcal{S}$**:

- **Terrain Map** (dim=($16 \times 16 \times 2$) with granularities of $[1.0, 2.0]$): The relative terrain waypoints in the agent's surrounding area. Various granularities capture different details and perceptual ranges effectively.

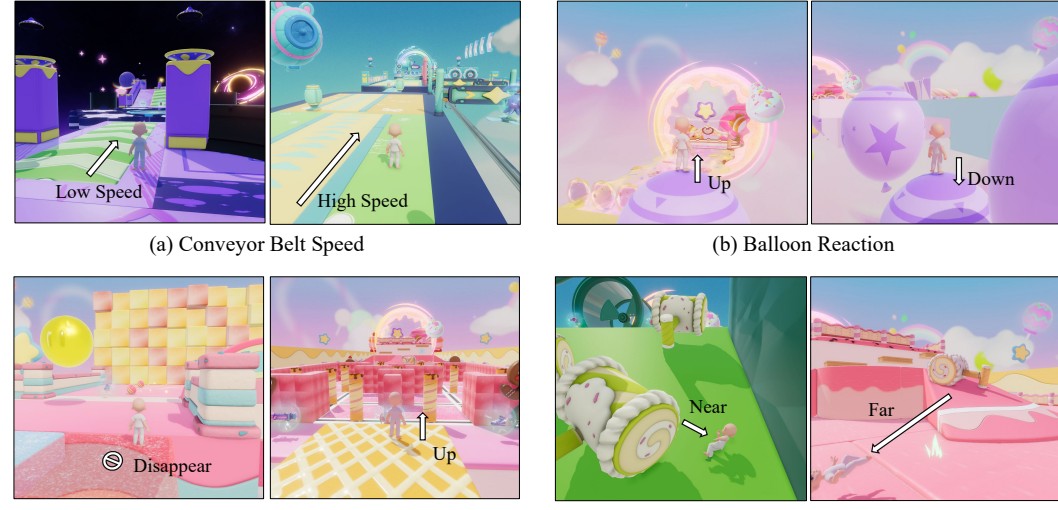

(a) Conveyor Belt Speed  (b) Balloon Reaction

(c) Floor Reaction  (d) Hammer Distance

Figure 7: More demonstrations of dynamics shifts in fall guys-like game. Colors and textures are only for visual enhancement and are not part of the agent's observations.

- **Item Map** (dim=$(16 \times 16 \times 1)$ with granularities of $[1.0, 2.0]$): Map of nearby items or objects. Multiple maps focus on different item types, with granularities for varying spatial scales.
- **Target Map** (dim=$(16 \times 16 \times 1)$ with granularities of $[1.0, 4.0, 16.0]$): Map of archive points locations. Various granularities capture different details and perceptual ranges at different spatial scales.
- **Goal Map** (dim=$(16 \times 16 \times 1)$ with granularities of $[1.0, 4.0, 16.0]$): Map of intermediate goal locations. Various granularities capture different details and perceptual ranges at various spatial scales.
- **Agent Info** (dim=32): Details about agent's own state, including position, rotation, velocity, animation state, and forward direction.
- **Destination Info** (dim=9): Details about the destination, including position, rotation, and size of the destination, providing crucial details for navigation and goal achievement.

**Action space** $\mathcal{A}$:

- **MoveX** (dim=3): Move along the X-axis. The three discrete options typically represent movement in the positive X direction, negative X direction, or no movement.
- **MoveY** (dim=3): Move along the Y-axis, with three discrete options for movement in the positive Y direction, negative Y direction, or no movement.
- **Jump** (dim=2): Represents jump behavior. Options are to initiate a jump or not.
- **Sprint** (dim=2): Represents sprint behavior. Options are to start sprinting or not.
- **Attack** (dim=2): Executes an attack. The two discrete options are to initiate an attack or not.
- **UseProp** (dim=2): Utilizes a prop. The two discrete options indicate whether the prop is used or not.
- **UsePropDir** (dim=8): Determines the direction for prop usage. The eight discrete options offer various directional choices for prop utilization.
- **Idle** (dim=2): Represents idle behavior. Options are to remain idle or not.

**Reward** $r$:

- **Arrive Target** (value=1.0): Rewards the agent for successfully reaching the archive point, with a positive reward of 1.0 upon achievement.

- **Arrive Goal** (value=0.3): Rewards the agent for reaching intermediate goal locations within the environment, with a positive reward of 0.3.

- **Arrive Destination** (value=1.0): Rewards the agent for reaching the final destination or endpoint within the environment, motivating task completion.

- **Goal Distance** (decay rate=0.05): Offers distance-based rewards, varying based on proximity to specific goal locations. Rewards diminish as the agent moves away from the goal, with distinct values for different distance ranges.

- **Fall or Wall** (value=$-1.0$): Penalizes the agent for continuously hitting the wall or falling off a cliff with a penalty of -1.0.

- **Stay** (value=$-0.01$): Penalizes the agent for remaining stationary for extended periods, encouraging continuous exploration and movement.

- **Time** (value=$-0.02$): Penalizes each time step, encouraging efficient decision-making and timely task completion.

**Network Architecture**    The network architecture is structured as follows: The Terrain Map, Item Map, Target Map, and Goal Map are each fed into a convolutional neural network (CNN) with ReLU non-linearity, followed by a fully connected network (FCN). This process yields four separate 32-dimensional vector representations for each respective map. The Destination Info and Agent Info are independently input into attention layers, generating 32-dimensional vectors for each. Subsequently, all 32-dimensional vectors (from the CNNs and attention layers) are concatenated into a single feature vector. The concatenated feature vector undergoes processing by a multi-head FCN to yield various output actions. Additionally, the concatenated feature vector is processed by another FCN to produce a value as the value function estimator.

**Training Setup**    We utilized the Ray RLlib framework (Liang et al., 2018), configuring 100 training workers and 20 evaluation workers. The batch size was set to 1024, with an initial learning rate of $1 \times 10^{-3}$, which linearly decayed to $3 \times 10^{-4}$ over 250 steps. An entropy regularization coefficient of 0.003 was employed to ensure adequate exploration during training. The training was conducted using NVIDIA TESLA V100 GPUs and takes around 20 hours to train 6M steps.

