# OpenReview forum: "ASOR: Anchor State Oriented Regularization for Policy Optimization under Dynamics Shift"
_ICLR.cc/2025/Conference — Submitted to ICLR 2025_

### Official Review · Reviewer_gtbd · 2024-10-27

**Soundness:** 2
**Presentation:** 2
**Contribution:** 2
**Rating:** 3
**Confidence:** 3

**Summary:**

The imitation from observation approaches are built on the assumption that stationary state distributions induced by the optimal policies are similar. However, under the dynamic shift of the environments, this assumption is normally violated. The paper addresses this issue by proposing the concept of anchor states and incorporating the anchor state distributions into the policy regularization. Some theoretical results are also provided, trying to provide some lower-bound guarantee. Finally, the proposed method is demonstrated on several common environments.

**Strengths:**

The proposed method seems technically sound.

The experiments consider most of the commonly used benchmarks, which seem extensible.

**Weaknesses:**

**1. Motivation.** It is not clear whether the violation of the similar stationary state distributions for imitation from observation approaches is critical for performance. No empirical evidence has been provided, so it is hard to tell whether this paper is well-motivated or not.

**2. Methodology.** As far as I can tell, the main idea for addressing the dynamic shift is to focus on the shared accessible states across all environments, defined as the so-called anchor states. However, I could not figure out the rationale for such an idea. Is the inaccessible state detrimental to policy learning and its generalization across all dynamics? I am very confused about the motivation behind it.

**3. Scope of Dynamic shift.** This paper only considers the Hidden-parameter MDP, which significantly limits the scope of dynamic shifts to be considered. My feeling is that although different parameters can be initialized by drawing samples from a certain distribution, the classes of MDP typically share similar structures or have a large overlap with each other. I could not tell the reward function needs to remain the same, until I find the authors state this point in the end. I think this limitation should be clearly stated at the beginning. That being said, the proposed method can only consider the a potentially small class of transition dynamic shift (by Hidden parameter MDP), while failing to incorporate the shift in the reward function.


**4. The theoretical results are limited and questionable.** Firstly, what is the relationship between Theorems 2 and 4? It is less convincing to consider the neural distance, which is typically considered in generative models. After I read the proof in the appendix, I found the proof highly relies on the assumption from Xu et al. 2020, assuming the reward function has a linear space structure. This assumption has not been clearly emphasized in Theorem 4.4 since the proof in the appendix highly depends on the proof from Xu et al. 2020. Thus, I do not think the theoretical results are sufficient to provide insightful results.

Another point that I am concerned about is the linear horizon dependency, which suggests the improved lower bound by authors. There is no direct evidence to demonstrate, or at least empirically, that the derived bounds are sharp or not vacuous to align with practice. The common RL theory normally promotes some regret analysis, sample complexity analysis, or small-error bound. However, it is not clear the linear horizon dependency (compared with quadratic one) can be widely accepted. The authors could provide some papers that also show a similar proof technique. I am not sure about this point, so feel free to correct me if I am wrong.


**5. Clarity and Writing.** After reading this paper, I am afraid I can not understand the logic behind it. What critical issues is this paper really focused on addressing? Is this issue really important in the literature? The clarity needs to be significantly improved. Some statements are confusing, such as KL divergence, which has a poor generalization (in Line 318). The author should be very careful about this statement.

**Minors**. How do you select the best \pho in Line 270? The experiments look fine to me, although I do not think there is any notable conclusion I can find. The paper has a large overlap with [1], which affects the novelty and technical contribution of this paper. Some detailed comparisons are also needed in the future.


[1] Zhenghai Xue, Qingpeng Cai, Shuchang Liu, Dong Zheng, Peng Jiang, Kun Gai, and Bo An. State regularized policy optimization on data with dynamics shift. In NeurIPS, 2023a.

**Questions:**

Please see the weakness.

---

> ### Author Response · Authors · 2024-11-20
> **Author Response**
>
> We thank the reviewer for the positive and constructive feedback. We are encouraged that the reviewer acknowledged the technical soundness and the extensibility of the proposed algorithm. With respect to weaknesses and questions, we provide our responses as follows.
>
> **Q1: Motivation of the Methodology**
>
> A: The reviewer raised concerns on whether inaccessible states and different optimal state distributions are detrimental to policy learning. One direct outcome of inaccessible states is that the optimal trajectory in one dynamics will no longer be optimal in the other. Imitating states in such a trajectory may lead to degraded overall performance.
>
> In the lava world example with dynamics shift in Sec. 3.1 and Fig. 1, if the policy learns from the optimal state trajectory in the upper environment, it will take more steps and receive more negative rewards to reach the goal compared with the optimal policy. Another example is autonomous driving in different traffic densities, as illustrated in Appendix B and Fig. 6. The optimal state trajectory under low traffic densities will include many states with high speeds. But  policies regularized by such state distribution will be more likely to crash under high traffic densities and will not be optimal.
>
> With regard to empirical evidences, baseline algorithms BCO, SOIL, and SRPO are motivated by the idea of similar stationary state distributions, which all show inferior performances compared with AGSA. Among these, SRPO follows a similar pipeline of reward augmentation, but still performs worse than AGSA in all involved experiments. We hope such discussions on the motivation of the methodology can convince the reviewer on the importance of introducing anchor states and performing policy regularization upon them.
>
> **Q2: Limited scope of dynamics shift**
>
> A: The reviewer raised concerns on the limited scope of MDPs with similar structures and failing to incorporate shift in the reward function. We discuss these two concerns respectively and will add the discussion to the revision.
> - **Dynamics Shift:** It is true that this paper relies on similar structures of MDPs. But without such similar structures, data from one MDP would never be informative for training policies in other MDPs. The only choice would be to train separate policies in each MDPs. One of the contributions of this paper is that we both motivate from (Sec. 3.2) and formally define (Def. 4.1) the similar structures, exploit them in policy optimization, and design a new training algorithm that improves data efficiency.
>
>   In fact, existing literature more or less rely on assumptions on the MDP structures. Off-dynamics learning methods, such as DARC [1], DARA [2], and H2O [3], are also built upon HiP-MDP and further assume access to the data distribution of the test environments. Our method is free of such additional information while achieving superior performance. SRPO [4] and Imitation from Observation methods [5,6] has a shared research scope with our paper, but is built upon a stronger assumption on the MDPs.
>
>   We would also like to mention that the setting of HiP-MDP is already general enough to handle a decent number of real-world senarios, such as autonomous driving with different road, tyre, weather, and traffic conditions, recommender systems with different news trend, user behaviors, and user preferences, video games with different map components, obstacle layouts, and so on. All of these tasks contain similar MDP structures that can be utilized in training.
> - **Reward Function:** We would like to clarify that dealing with shifts in reward functions will be a complete different research problem. To our best knowledge, none of the existing literature for dynamics shift [1,2,3,4,5,6] considers potential reward shifts. To handle reward shifts, a typical approach will be meta-learning that aims at fast test-time adaptation with few-shot samples, which will be out of the scope of this paper.
>
> **Q3: Clarity and writing**
>
> A: The critical issue we focus is discussed in the first paragraph of the paper (Lines 31-41) and briefly stated in lines 40-41: *how can RL policies be efficiently optimized using data collected under dynamics shift?*. It is formalized in Sec. 2.1 (lines 93-94): *Policy optimization under dynamics shift aims at finding the optimal policy that maximizes the expected return under all possible* $\theta\in\Theta$. Such issue is important not only in literature, but also in many practical scenarios. There are quiet a number of related literature in this field as discussed in Sec 2.2 (Related Work). Real-world applications, such as autonomous driving, video games, robotics locomotion, and recommendation systems, may encounter dynamics shift in the training environment, which traditional RL algorithms struggle to deal with. So it is vital to design more robust and efficient training algorithms.

---

> ### Author Response · Authors · 2024-11-20
> **Author Response**
>
> **Q4: Limited and questionable theoretical results**
>
> A: We appreciate that the reviewer takes the precious time to read the theoretical results and the proofs. The reviewer's concerns are discussed as follows.
> - **Relationship between Thm. 4.2 and 4.4**: To analyze the KL-contrained optimization problem, Thm. 4.2 assumes that the learning policy $\hat\pi$ minimizes the KL divergence in Eq. (1) to be smaller than $\varepsilon$. But in practice, we can only minimize the divergence between the *empirical* version of distributions on finite training data. It remains unclear that given finite samples, how close the divergence between *real* distributions will be with high probabilities and how will such divergence influence the final performance lower bound. Thm. 4.4 gives such finite-sample analysis by leveraging the generalization ability of the network distance.
> - **Network distance and KL Divergence**: The network distance has been proved to have good generalization property in finite sample analysis, where the intuitively preferred KL divergence fails to provide such property [Arora et al., 2017; Xu et al., 2020]. So it is natural to consider the network distance to derive the finite sample analysis of Thm. 4.4. This will surely introduce divergence from the practical algorithm, but we think it is a necessary trade-off to get meaningful theoretical results.
> - **Assumption from [Xu et al., 2020]**: We actually emphasized the reliance on the linear structure of reward function by writing
> > if the reward function $r_{\hat\pi,T}(s)=\mathbb E_{a\sim\hat\pi,s'\sim T} r(s,a,s')$ lies in the linear span of $\mathcal P$
>
>    in Thm. 4.4. It is true that our paper borrow some proving techniques for finite sample analysis from Xu et al. (which has been clearly stated in the appendix), but these two papers work on completely different problems. Xu et al. focuses on Imitation Learning with Generative Adversarial Imitation Learning (GAIL), while our paper focuses on policy optimization under dynamics shift. To our best knowledge, among existing literature for policy optimization under dynamics shift, our paper is the first to discriminate empirical and real distributions and give finite sample analysis. So the insight here is that a PAC bound like Eq. (6) can be proved for policies regularized by the network distance. Future work that involves a similar procedure may also derive a PAC bound and compare with ours.
> - **Linear horizon dependency:** To the best of our knowledge, Xu et al. [9] is the first to point out the advantage of linear horizon dependency over quadratic one and use it to explain GAIL's superior performance over behavior cloning. The idea behind this advantage is simple: with different horizon dependencies, the imitation error will accumulate at different rates along the trajectory. Before and after that, several papers proposed **quadratic bounds** when analyzing performance differences in face of dynamics or policy divergence. The coefficients of these bounds vary across different problem formulations. MBPO [7] consider model-based RL with learned dynamics model that has a gap of $\varepsilon_m$ with the ground truth dynamics, leading to the following bound with quadratic horizon dependency.
> $$
> \eta[\pi] \geq \hat{\eta}[\pi]-[\frac{2 \gamma r_{\max }(\epsilon_m+2 \epsilon_\pi)}{(1-\gamma)^2}+\frac{4 r_{\max } \epsilon_\pi}{(1-\gamma)}]
> $$
>    SLBO [8] also focuses on model-based RL and gives the following bound with quadratic horizon dependency:
> $$
> \left|V^{\pi, \hat{M}}-V^{\pi, M^{\star}}\right| \leq \kappa L \underset{\substack{S \sim \sim \pi_{\text {ref }} \\ A \sim \pi(\cdot \mid S)}}{\mathbb{E}}[\|\hat{M}(S, A)-M^{\star}(S, A)\|]+2 \kappa^2 \delta B,
> $$
>   where $\kappa=\gamma(1-\gamma)^{-1}$. EGPO [10] considers the shared autonomy of a learning controller and a expert controller, proposing that the value difference $\hat V$ can be bounded as follows:
> $$
> \hat{V} \leq \frac{\epsilon}{1-\gamma}(1+\frac{1}{\eta}+\frac{\gamma}{1-\gamma} K_\eta^{\prime}),
> $$
>    where $K_\eta'$ describes the policy divergence between two controllers and its coefficient also has quadratic horizon dependency. Due to the notoriously high rate in error accumulation of quadratic horizon dependency, the aforementioned papers come up with special methods for error reduction. For example, MBPO propose to rollout with the learned dynamics model for no longer than K steps, rather than the whole trajectory. EGPO propose to involve human participants to select a few steps that the expert controller should take over.
>
>    In this paper, we formalize and utilize the similar structures of MDPs with dynamics shift and derive a new performance bound with linear dependency. Thanks to this good property, we are able to design policy regularization on the stationary state distribution, which is computed along whole trajectories. We hope such brief literature review could inform the reviewer of the importance of linear horizon dependency.

---

> ### Author Response · Authors · 2024-11-20
> **Author Response**
>
> **Q5: How to select $\rho$ in line 270?**
>
> A: $\rho_1$ and $\rho_2$ are selected through a few rounds of hyperparameter tuning. Their effects are examined in the ablation studies in Table 2, where ASOR shows robustness to hyperparameter changes.
>
> **Q6: Conclusion from the experiments**
>
> A: As stated in Sec. 5 (Lines 360-365), we conduct experiments to investigate whether ASOR can efficiently learn from data with dynamics shift and can be a general add-on module to different algorithms in diverse application scenarios. With empirical results in four different RL tasks, we show that ASOR can be effectively integrated with several state-of-the-art
> cross-domain policy transfer algorithms and substantially enhance their performance.
>
> **Q7: Novelty over a related work [4]**
>
> A: SRPO [4] focus on a similar setting with this paper, but is based on a strong assumption of universal identical state accessibility. We demonstrate that such an assumption will not hold in many tasks and a more delicate characterization of state accessibility will lead to better theoretical and empirical results.
>
> **References**
>
> [1] Off-Dynamics Reinforcement Learning: Training for Transfer with Domain Classifiers. ICLR 2021.
>
> [2] DARA: Dynamics-Aware Reward Augmentation in Offline Reinforcement Learning. ICLR 2022.
>
> [3] When to Trust Your Simulator: Dynamics-Aware Hybrid Offline-and-Online Reinforcement Learning. NeurIPS 2022.
>
> [4] State Regularized Policy Optimization on Data with Dynamics Shift. NeurIPS 2023.
>
> [5] State-only imitation with transition dynamics mismatch. ICLR 2020.
>
> [6] Offline imitation learning with a misspecified simulator. NeurIPS 2020.
>
> [7] When to Trust Your Model: Model-Based Policy Optimization. NeurIPS 2019.
>
> [8] Algorithmic Framework for Model-based Deep Reinforcement Learning with Theoretical Guarantees. ICLR 2019.
>
> [9] Error Bounds of Imitating Policies and Environments. NeurIPS 2020.
>
> [10] Safe Driving via Expert Guided Policy Optimization. CoRL 2021.

---

> ### Author Response · Authors · 2024-11-25
> **Comments are Welcomed**
>
> We sincerely value your dedicated guidance in helping us enhance our work. We are eager to ascertain whether our responses adequately address your primary concerns, particularly in relation to the motivation, the scope of dynamics shift, and the theoretical results. We would be grateful for the opportunity to provide any needed further feedback.

---

> > ### Comment · Reviewer_gtbd · 2024-11-25
> >
> > Thanks for the authors' response and I acknowledge that I have read the authors' response. However, my impression on this work remains the same. For example, this paper looks more like an incremental work over [4]. Therefore, I keep my rating for now.

---

### Official Review · Reviewer_sFee · 2024-10-31

**Soundness:** 1
**Presentation:** 2
**Contribution:** 1
**Rating:** 3
**Confidence:** 4

**Summary:**

This paper investigates a problem in Imitation from Observation (IfO) that occurs when certain states become inaccessible due to changes in the dynamics between the expert and the agent. The authors introduce a reward augmentation algorithm called ASOR (Anchor State Oriented Regularization) to tackle this issue. ASOR works by excluding reference states that change in accessibility during training and by defining anchor states that remain accessible across all dynamics.

**Strengths:**

The paper points out a challenge in cross-domain imitation learning from observation, where some actions in the expert demonstrations are not realizable by the agent.

**Weaknesses:**

1.	The motivation of the paper is not entirely convincing. Even in in-domain imitation from observation—where the experts and agent operate in the same environment—optimal policy trajectories are often diverse due to randomness or differing initial conditions. As a result, the state sets of these trajectories are unlikely to fully overlap; therefore, most of these states won't fit within the anchor states introduced in the paper, which are indeed realizable by the agent.
2.	The empirical results show limited improvement over baseline methods, suggesting that the proposed ASOR algorithm may not offer a significant performance advantage.

**Questions:**

The core assumption behind ASOR seems to be that anchor states exist and can be consistently identified across environments. However, in many real-world settings, this assumption seems unrealistic (see weakness 1). Is there a practical strategy for dynamically identifying anchor states in such complex environments? The current approach assumes these states remain fixed across all trajectories, which could be overly simplistic.

---

> ### Author Response · Authors · 2024-11-20
> **Author Response**
>
> We thank the reviewer for the constructive feedback. We provide our responses to reviewer's concerns as follows.
>
> **Q1: The motivation of the paper is not entirely convincing.**
>
> A: We refer the reviewer to the general response for discussions on the motivation of anchor states. In general, the practical algorithm in Sec. 3.4 design an approximated approach of identifying anchor states in continuous state spaces, where the exact anchor state will not exist.
>
> **Q2: The empirical results show limited improvement over baseline methods**
>
> A: We would like to clarify that our AGSA algorithm is evaluated as an add-on module to existing algorithms. It requires minimal adjustment to the original training pipeline, which is the reward augmentation with two density ratios. The improvement over base algorithms is much higher than other algorithms as add-on modules. Take the performance in D4RL tasks as an example. The DARA algorithm only improves the performance of the MAPLE algorithm by an average of 2.7%. SRPO can improve by 11.1%, while our AGSA algorithm can improve by 25%.
> AGSA is also general enough to be applicable to five different training tasks, including Gridworld, D4RL, MuJoCo, MetaDrive, and a large-scale video game, achieving constant performance improvements. So it may be unfair to conclude that the empirical improvement is limited.
>
> **Q3: Practical strategy for dynamically identifying anchor states**
>
> A: Actually we have already employed a practical algorithm for identifying a "soft version" of anchor states in continuous state spaces. We refer the reviewer to the general response for detailed discussions. Four out of the five involved training environments have continuous state spaces. As discussed in Q2, the practical approximation shows decent performance improvements in these environments.

---

> ### Author Response · Authors · 2024-11-25
> **Comments are Welcomed**
>
> We sincerely value your dedicated guidance in helping us enhance our work. We are eager to ascertain whether our responses adequately address your primary concerns, particularly in relation to the motivation and the empirical results. We would be grateful for the opportunity to provide any needed further feedback.

---

### Official Review · Reviewer_zXdd · 2024-11-01

**Soundness:** 2
**Presentation:** 2
**Contribution:** 3
**Rating:** 5
**Confidence:** 3

**Summary:**

This paper recognizes a limitation in existing RL under the dynamics shift method. They assume stationary state distributions induced by optimal policies remain similar despite the dynamics shift. This limitation does exist in many off-dynamics RL/RL under dynamics mismatch. Thus, they propose a regularized policy optimization algorithm that regularizes the 'anchor state' instead of the 'all-state' stationary distribution. The anchor state is defined as the accessible state across all the optimal trajectories in all the MDP.  Further, they provide a lower bound for the performance.

**Strengths:**

1, The method is well-motivated by the limitation of existing RL under dynamics shift. And regularizing the 'anchor state' distribution is novel.

2, The experiments is comprehensive, demonstrating the effectiveness of the method.

**Weaknesses:**

1, the problem is not formally defined. Can you be more specific about policy optimization under dynamics shift? Some paper, such as off-dynamics RL, focus on training a policy in the source (simulation) environment(s) and deploying to the target (real world) environment. This can also be called as policy optimization under dynamics shift. It looks like the focus is on training a policy on multiple MDPs with different transitions and performing simultaneously well in all the environments.

2, section 3.3 and 3.4 is hard to follow. How to connect eq(3) with the eq(2) and how do you get eq(3) with Random Network Distillation (RND). What is the loss function?  Also, more details of ESCP should be included, and the notation of the ESCP should be defined formally instead of just mentioning it in the algorithm 1. And the notation is a bit complicated.

**Questions:**

1, What if the anchor state doesn't exist? Is there any assumption you need? It can be the case that the optimal trajectories in different MDP won't intersect or the intersection state is limited.

2, See weakness 1(b) about section 3.2.

3, Why using an arbitrary $T_0$ in the loss, how do you choose $T_0$ how it affect the results? Does the anchor state in $T_0$ cover all the anchor states across different MDPs? If yes, can you justify it? If not, the the Eq(2) seems different from the motivation.

4, how do you get the anchor state during training?

I would be happy to raise my score if the questions are answered.

---

> ### Author Response · Authors · 2024-11-20
> **Author Response**
>
> We thank the reviewer for the in-depth and constructive feedback. We are encouraged that the reviewer acknowledged the clear motivation, novel regularization method, and comprehensive experiments. With respect to weaknesses and questions, we provide our responses as follows.
>
> **Q1: Problem formulation**
>
> A: The reviewer's speculation on the focus is correct. In our HiP-MDP formulation in Section 2.1 (Lines 93-94), policy optimization under dynamics shift aims at finding the optimal policy that maximizes the expected return under all $\theta\in\Theta$, i.e., all possible dynamics in the HiP-MDP. In the revision we will make it a formal definition and define the training environments, which have hidden parameters $\theta_1,\theta_2,\cdots,\theta_n\in\Theta$. Compared with the formulation of off-dynamics RL, our HiP-MDP formulation does not assume any prior knowledge to target environments, which can be a natural fit to tasks with highly dynamic and non-stationary environments, such as robotics locomotion in open environments, autonomous driving, video games with evolving map components, and recommender systems.
>
> **Q2: More details in Section 3.3 and 3.4**
>
> A: We apologize for the obscure statements in Section 3.3 and 3.4. We give more clear demonstrations as follows which hopefully make our ideas easier to follow. They will also be added to the revision if the reviewer finds helpful.
> - **Connection with Eq. (3) and Eq. (2)**: Sorry for the redundant subscript t in the original Eq. (3). The density ratio $\frac{d_{T_0}^{\*,+}\left(s\right)}{d_T^\pi(s)}$ in Eq. (3) is the first reward augmentation term in Eq. (2). Eq. (3) breaks down the density ratio $\frac{d_{T_0}^{\*,+}\left(s\right)}{d_T^\pi(s)}$ in Eq. (2) into two density ratios that are easier to compute.
> - **Getting Eq. (3) with RND**: According to Lemma 3.2 (Lines 209-213), the density ratio $\frac{d_{T_0}^{\pi,+}(s)}{d_T^\pi(s)}$ can be obtained by maximizing $
> \mathbb{E}\_P[f^{\prime}(\omega(s))]-\mathbb{E}\_Q[f^{\*}(f^{\prime}(\omega(s)))] (\*),
> $
> where $f$ is a predefined function, $P$ and $Q$ are cumulative probability distributions of the density functions $d_{T_0}^{\pi,+}(\cdot)$ and $d_T^\pi(\cdot)$. Therefore, to obtain the density ratio, we only need to draw samples from $d_{T_0}^{\pi,+}(\cdot)$ and $d_T^\pi(\cdot)$, and then maximize Eq. (\*). As $d_{T_0}^{\pi,+}(\cdot)$ is the distribution of anchor states visited by all optimal policies under different dynamics, states that are sampled from it should have higher visitation frequency. We therefore select RND as one of the visitation measure in large-scale tasks with continuous state spaces. Its loss is $L\_{\text{RND}}=\mathbb E\_{s\in\mathcal R}\|f\_0(s)-f\_\theta(s)\|\_2$, where $\mathcal R$ is the replay buffer, $f_0$ is a randomly initialized network, and $f_\theta$ is the learning network with parameter $\theta$. States with smaller RND loss are regarded to have higher visitation counts and more likely to be sampled from $d_{T_0}^{\pi,+}(\cdot)$.
> - **Details of ESCP**: The Environment Sensitive Contextual Policy learning (ESCP) method trains an RNN-based context encoder $\phi(z_t|s_t,a_{t-1},z_{t-1})$, where $z_t$ and $z_{t-1}$ are latent variables that are trained to align with the environment parameter. By training policies $\pi(a|s_t,z_t)$ conditioned on latent variables, ESCP has the ability to adapt to environments with dynamics shift. Our AGSA method aims at efficiently exploiting data collected under dynamics shift. AGSA can work as an add-on module over ESCP and substantially enhances ESCP's performance. We will include more details of ESCP as well as its notations in the revision.
>
> **Q3: Existence of anchor states and getting anchor states during training**
>
> A: We refer the reviewer to the general response for discussions on anchor states. In general, the practical algorithm in Sec. 3.4 design an approximated approach of identifying anchor states in continuous state spaces, where the exact anchor state will not exist.

---

> ### Author Response · Authors · 2024-11-20
> **Author Response**
>
> **Q4: Arbitrary $T_0$ in the loss**
>
> A: The advantage of an arbitrary $T_0$ in the loss is that we do not have to care about the specific dynamics that the optimal policy is in. In the proposed AGSA algorithm, the optimal state distributions from *any* dynamics can contribute to policy learning in new dynamics. This greatly increases the data efficiency of AGSA as shown in the empirical results. In practice, we do not need to choose a specific $T_0$, since an arbitrary $T_0$ fits in the optimization objective Eq. (2). We just mix all training data collected from environments with different dynamics. According to Eq. (3), any states with high state value or frequently visited by the learning policy can be used to obtain the density ratio $\frac{d\_{T_0}^{\*,+}\left(s\right)}{d\_T^\pi(s)}$ and $\frac{d\_{T_0}^{\*,-}\left(s\right)}{d\_T^\pi(s)}$ in AGSA's reward augmentation pipeline.
>
> With respect to the concern on anchor states in $T_0$, $T_0$ can indeed cover all anchor states across different MDPs. This is because in Def. 3.1, an anchor state $s$ should have non-zero visitation count in all dynamics, including an arbitrary dynamics $T_0$. Intuitively, the anchor states are the intersection of all optimal state trajectories under different dynamics and will therefore be a subset of the states reachable by $T_0$.

---

> ### Author Response · Authors · 2024-11-25
> **Comments are Welcomed**
>
> We sincerely value your dedicated guidance in helping us enhance our work. We are eager to ascertain whether our responses adequately address your primary concerns, particularly in relation to the existence of anchor states, the arbitrary $T_0$ in the loss, and details in practical approximations. We would be grateful for the opportunity to provide any needed further feedback.

---

### Official Review · Reviewer_TJem · 2024-11-03

**Soundness:** 4
**Presentation:** 2
**Contribution:** 4
**Rating:** 8
**Confidence:** 3

**Summary:**

The paper introduces ASOR (Anchor State Oriented Regularization), a novel approach for policy optimization in reinforcement learning under dynamics shift. In real-world scenarios, environmental dynamics can change significantly, making certain states inaccessible and rendering traditional Imitation from Observation (IfO) techniques, which assume identical optimal state distributions, ineffective. ASOR addresses this by focusing on "anchor states"—states accessible across all dynamics—which it incorporates into policy regularization to improve robustness and adaptability. The paper also demostrates the versatility of the method as an add-on for state-of-the-art policy optimization methods across different settings. ASOR is validated across various online and offline RL benchmarks, demonstrating its efficacy in scenarios with dynamic changes.

**Strengths:**

**Strengths**

- Conceptual Intuition and Motivation: The paper introduces a method with strong intuitive appeal by addressing the limitations of state-only policy transfer algorithms, which often assume identical state distributions across environments. The authors effectively motivate the need for a refined approach, introducing anchor states as a way to target consistently accessible states. This innovation weakens the assumption of identical state distributions and proposes a structured method to identify and match distributions on anchor states, which is a valuable contribution.
- Theoretical Analysis: I didn't follow all the theoretical analysis. But the paper claims to be obtaining stronger performance lower bounds while making weaker assumptions compared to the baselines.
- Compatibility with Existing Algorithms: ASOR is modular and integrates smoothly into existing algorithms by modifying the reward function. The paper demonstrates ASOR’s effectiveness as an enhancement to state-of-the-art RL algorithms like PPO, MAPLE, and ESCP across different RL settings, which increases its potential applicability.
- Empirical Validation Across Diverse Environments: ASOR is validated across a range of environments, including Minigrid, D4RL, MetaDrive, and a high-dimensional Fall Guys-like game, covering both online and offline RL settings. The method consistently improves performance, showcasing ASOR’s versatility and impact across environments with dynamics shifts.

**Weaknesses:**

**Weaknesses**
- Writing and Background Accessibility: At points, the writing was challenging to follow for me. This may be partly due to my fading memory of some of the background needed for the work. Particularly in Section 3.3, which references previous work with minimal explanation.  Including foundational information or more detailed background in the appendix would make the paper more accessible, especially to readers less familiar with some of these topics.
- Anchor State Identification: Anchor state identification is central to the  approach, yet it is only briefly discussed in Section 3.3. A deeper examination of the considerations, challenges, and potential tradeoffs in anchor state identification would strengthen the method’s presentation and provide clarity on this crucial component.
- Limited Ablations on Policy Regularization Choices: While it makes sense to regularize policies to maintain consistent distributions on anchor states, the decision to explicitly incentivize divergence on non-anchor states is less straightforward. This is particularly surprising given that the anchor states are identified using learned policy distributions and state values. More ablations or additional exposition on the rationale behind this choice would provide valuable insights and a clearer understanding of the intended impact of this aspect of the method.

**Questions:**

same as weaknesses

---

> ### Author Response · Authors · 2024-11-20
> **Author Rebuttal**
>
> We thank the reviewer for the positive and constructive feedback. We are encouraged that the reviewer acknowledged the clear and effective motivation, the theoretical analysis, the compatibility with existing algorithms, and the empirical validations of AGSA. With respect to weaknesses and questions, we provide our responses as follows.
>
> **Q1: Writing and Background Accessibility**
>
> A: We thank the reviewer for the tolerance of our explanations in the current version. In general, Sec. 3.3 introduces how we approximate the density ratios $\frac{d_{T_0}^{\*,+}\left(s_t\right)}{d_T^\pi\left(s_t\right)}$ and $\frac{d_{T_0}^{\*,-}\left(s_t\right)}{d_T^\pi\left(s_t\right)}$ in practice. Lem. 3.2 basically turns the density ratios into the argmax of an optimization problem, which requires to sample from the distributions $d_{T_0}^{\*,+}(\cdot)$,  $d_{T_0}^{\*,-}(\cdot)$, and $d_T^\pi\left(\cdot\right)$. The rest of Sec. 3.3 discusses how to approximately sample from these distributions when exact anchor state distribution is unavailable. We will add more explanations on Lem. 3.2 and the binary observation state $\mathcal O_t$ in the appendix of the revised paper.
>
> **Q2: Anchor State Identification**
>
> A: We refer the reviewer to the general response. We discuss how we design the approximation on anchor states in Sec. 3.3, where exact anchor state distributions are unavailable.
>
> **Q3: Incentivizing divergence on non-anchor states**
>
> A: We conduct ablation studies for encouraging divergence on non-anchor states in the large-scale video game environment. According to the results in the following table, AGSA without considering non-anchor states has lower overall performance than the original AGSA. This can be attributed to its high trapped rate and low policy entropy. The trapped rate indicates how frequent the agent gets stuck and stand still until the end of the episode. The policy entropy denotes the diversity of the agent behavior. These two metrics indicate a poor exploration ability without divergence maximization, which also demonstrates the effectiveness of encouraging divergence on non-anchor states.
>
> |                                    | Total Reward (↑) | Goal Reward (↑) | Success Rate (↑) | Trapped Rate (↓) | Unnecessary Jump Rate (↓) | Distance from Cliff (~) | Policy Entropy (~) |
> | ---------------------------------- | ---------------- | --------------- | ---------------- | ---------------- | ------------------------- | ----------------------- | ------------------ |
> | **PPO**                            | 0.329±0.308      | 1.154±0.085     | 0.361±0.009      | 0.012±0.009      | 0.064±0.003               | 0.152±0.025             | 5.725±0.185        |
> | **PPO+SRPO**                       | 0.337±0.257      | 1.513±0.076     | 0.376±0.006      | 0.038±0.015      | 0.040±0.003               | 0.148±0.018             | 5.859±0.098        |
> | **PPO+AGSA**                       | **0.554**±0.336  | **1.781**±0.053 | **0.387**±0.005  | **0.005**±0.005  | 0.029±0.003               | 0.143±0.021             | 6.358±0.122        |
> | **PPO+AGSA w\o Non-anchor states** | 0.519±0.292      | 1.637±0.049     | 0.382±0.006      | 0.019±0.013      | **0.027**±0.005           | 0.145±0.025             | 5.633±0.110        |

---

### Author Response · Authors · 2024-11-20
**General Author Response**

We thank all reviewers for their in-depth and constructive comments. Some reviewers raise concerns on our motivation of anchor states. Reviewers zXdd and sFee are concerned that anchor states may not exist. Reviewers zXdd and Tjem are interested in more details in identifying anchor states. Reviewer sFee considers practical strategies for anchor state identification. In general, our paper indeed follows a pipeline from motivation to practice for identifying and utilizing anchor states. The idea is motivated from certain tasks, and then approximated and generalized to broader and more complex scenarios. We provide detailed discussions on this pipeline in the bullet points and will add them in the revision.
- **Existence of anchor states:** Anchor states is defined in discrete state spaces with overlapped optimal state trajectories across different dynamics. This covers a decent number of tasks, such as gridworld, atari-like video games, board games, and text generation.
- **Practical approximation of anchor states:** Anchor states defined in Def. 3.1 will not exist in continuous state spaces, where the probability density on any specific states will be zero. We follow previous approaches [1,2] and turn to pseudo counts for identifying a "soft version" of anchor states. As discussed in Sec. 3.4 (Practical Algorithm), states with higher pseudo counts are more likely to be sampled from the anchor state distribution. They will be added to the buffers $D^+\_P$ and $D^+\_Q$ for training the density ratios $\omega^+$ and $\omega^-$, which are used in reward augmentation in the practical algorithm. The practical algorithm also involves a *relative* selection of anchor states, rather than *absolute* judgements. A fixed portion of states in the training batch will be regarded to be sampled from the anchor state distribution. This solves the concern of diverse optimal state trajectories that may not overlap. States that are visited by most (but not all) of the optimal policies may also be regarded as anchor states in practice.
- **Performance of Approximation**: We mainly focus on continuous state spaces in the experiments, which involve the aforementioned approximation of anchor states. Our ASOR algorithm can improve the performance of all selected base algorithms, including the on-policy PPO algorithm, the off-policy ESCP algorithm, and the offline MAPLE algorithm. This demonstrates the effectiveness of the approximation in practical scenarios.

**References**

[1] Count-Based Exploration with Neural Density Models. ICML 2017.

[2] Exploration by Random Network Distillation. ICLR 2019.

---

### Meta-Review · Area_Chair_X4DR · 2024-12-29

**Metareview:**

This paper introduces distribution constraints on the "anchor states" in policy for imitation learning under dynamic environment shift. The authors conducted empirical study on common benchmarks.

The major issues raised by the reviewers lies in following aspects:

1, The motivation of the regularization is not convincing. Although the authors illustrate the intuition in Fig 1, it still not clearly demonstrates the universality of the intuitiion in real-world applications.

2, The method is built on "anchor state", which, however, is not clearly discussed.

3, The derivation of the method and writing is not clear, and thus, making the paper difficult to follow.

In sum, I suggest the authors to consider the reviewers' comments to improve the paper.

**Additional Comments On Reviewer Discussion:**

During the rebuttal period, the authors provide additional experiments and more explanation, which partially address some concerns from the reviewers. However, most reviewers still remain their opinion.

---

### Decision · Program_Chairs · 2025-01-22

Reject